# On the use of discrete-time quantum walks in decision theory

**Ming Chen**[1]*, **Giuseppe M. Ferro**[1]*, **Didier Sornette**[1,2,3,4]*

**1** ETH Zürich, Department of Management, Technology and Economics, Zürich, Switzerland, **2** Institute of Risk Analysis, Prediction and Management, Academy for Advanced Interdisciplinary Studies, Southern University of Science and Technology, Shenzhen, China, **3** Tokyo Tech World Research Hub Initiative (WRHI), Institute of Innovative Research, Tokyo Institute of Technology, Tokyo, Japan, **4** Swiss Finance Institute, c/o University of Geneva, Geneva, Switzerland

* mingchen@ethz.ch (MC); ferrog@ethz.ch (GMF); dsornette@ethz.ch (DS)

**Data Availability Statement:** All relevant data are within the paper and its Supporting information files.

**Funding:** The authors received no specific funding for this work.

## Abstract

We present a short review of discrete-time quantum walks (DTQW) as a potentially useful and rich formalism to model human decision-making. We present a pedagogical introduction of the underlying formalism and main structural properties. We suggest that DTQW are particularly suitable for combining the two strands of literature on evidence accumulator models and on the quantum formalism of cognition. Due to the additional spin degree of freedom, models based on DTQW allow for a natural modeling of model choice and confidence rating in separate bases. Levels of introspection and self-assessment during choice deliberations can be modeled by the introduction of a probability for measurement of either position and/or spin of the DTQW, where each measurement act leads to a partial decoherence (corresponding to a step towards rationalization) of the deliberation process. We show how quantum walks predict observed probabilistic misperception like S-shaped subjective probability and conjunction fallacy. Our framework emphasizes the close relationship between response times and type of preferences and of responses. In particular, decision theories based on DTQW do not need to invoke two systems ("fast" and "slow") as in dual process theories. Within our DTQW framework, the two fast and slow systems are replaced by a single system, but with two types of self-assessment or introspection. The "thinking fast" regime is obtained with no or little self-assessment, while the "thinking slow" regime corresponds to a strong rate of self-assessment. We predict a trade-off between speed and accuracy, as empirically reported.

## Introduction

Researchers in many fields have tried to understand how humans make decisions in various tasks. In economics, scholars are interested in value-based decision-making, such as choosing one investment strategy or an insurance package. Similarly, cognitive and neurological scientists study how subjects make decisions in perceptual tasks, such as identifying the direction of moving objects or comparing the sizes of two objects. Although these two types of decisions in economics and in cognition appear different, it is often hypothesized that some parts of their

**Competing interests:** The authors have declared that no competing interests exist.

decision processes should be the same. After all, to judge whether a predator runs left or right while gathering food is a value-based decision with life-threatening risks in the wild.

The commonly implemented Expected Utility Theory [1] hypothesizes a decision-maker as an "omniscient, computationally omnipotent Economic Man" [2]. From the theory, a decision-maker (DM) is predicted to choose the alternative with the highest expected utility, formed by summing the values of events perceived by the DM (utility) weighted by the probabilities for the events to occur. However, the assumption of rationality is often violated (e.g. [3, 4]), making the theory more of a normative guidance instead of a descriptive model. In light of the descriptive inadequacy of Expected Utility Theory, two approaches, among others, have been shown to accurately describe human decision-making. The first class of models, known as accumulator models [5–7], represents the deliberation activity leading to a decision as a noisy accumulation of evidence over time. According to the models, choice is triggered once evidence accumulated in favor of one alternative reaches a given threshold. As for preferential choice under uncertainty, the most famous accumulator model is Decision Field Theory (DFT), where the decision process is represented by a (sophisticated) random walk [6]. Two of the present authors recently developed a model in the same fashion, called Stochastic Representation Decision Theory (SRDT [7]). Here, the choice set determines the topology of the space in which the representative random walk wanders. These models recognize fundamentally that decisions take time. As a result, they can account, among other patterns, for the role of time pressure on choice [8, 9]. Additionally, from the connectionist point of view, multiple evidence paths evolve in parallel and accumulate to construct a decision. These models can be also simplified in the form of classical random walks [10].

Notwithstanding the degree of sophistication achieved by classical theories of choice, they share the limitations imposed by the classical axioms of probability theory [11]. In particular, these theories cannot account for other prominent biases, such as order effects [12] and the conjunction fallacy [13].

Quantum formalisms of cognition on the other hand [14–17] propose that the mathematical properties of quantum probabilities are better suited to represent how human minds evaluate competing alternatives by accounting for superposition and entanglement effects. This provides a novel approach to account for both the probabilistic nature of decision-making and the interactions between prospects. The quantum formalism does not imply a physical quantum process in human brains; rather, it serves as a mathematical language to capture the properties associated with human decision-making (see sub-section "Decision theories based on static quantum models"). For example, quantum models provide a concise solution to the order effect. Due to non-commutativity of measurement with certain eigenbases, a quantum model naturally accounts for order effect, where one judgement can affect later judgements [18]. Also, the assumption in economics that choices reveal preferences and beliefs is challenged by several studies, suggesting that the process of decision contributes to the *construction* of preferences [19, 20]. In this respect, the active role of measurement in quantum models offers a natural resemblance to choices that shapes the preferences.

While resolving the anomalies and fallacies, quantum models appear to be more explanatory theories than predictive theories. One notable exception is Quantum Decision Theory (QDT) by Yukalov and Sornette [15]. QDT, besides rationalizing a large number of paradoxes [21], provides a parameter-free prediction known as the quarter-law [22], which has been shown to *quantitatively* account for observed choice patterns.

However, the underlying dynamics of a thinking process is ignored in static quantum models. Accumulator models, on the other hand, start by modeling plausible computational mechanisms represented by stochastic processes, which are taken as coarse-grained descriptions of the collective dynamics of more microscopic degrees of freedom (e.g. neurons or clusters of

neurons). The advantage of such models lies in their attempt to account even in some very rough way for the neurophysiologic substrate of decision activity, allowing to answer deeper questions, such as *why* people's choices are or appear to be stochastic.

Therefore, given the powerful characteristics of the quantum formalism of cognition on the one hand, and of noisy accumulator models on the other hand, we aim to explore the possible insights obtained by merging the two approaches, in a first attempt to formulate a computational theory of quantum cognition. The paper is constructed in the following way. In Section "Review of two branches of decision models", we review two branches of decision theories: evidence accumulation models and decision theories based on static quantum models. In sub-section "Previous decision theories based on quantum walks", we review the previous attempts to merge the two approaches, together with their limitations. Section "Framework of decision theory based on discrete-time quantum walks" introduces our general framework in terms of discrete-time quantum walks [23]. Section "Examples of application of the proposed quantum walk framework to decision-making" presents several novel models based on the framework that describe some prominent human biases. This serves as an illustration for the descriptive power of quantum walks. Section "Conclusion" concludes.

## Review of two branches of decision models

### Stochastic decision theories: Drift diffusion model, DFT, SRDT

Models of decision-making based on the idea of accumulation of noisy evidence over time have been very successful in different areas of cognition [5, 24, 25]. The choice process is described by a dynamic accumulation of evidence in favor of each possible action; the option whose cumulative evidence exceeds a threshold is chosen. Neuroscience research extensively supports such choice mechanism [26]. As our present focus is on preferential choice under uncertainty, we will summarize how decision field theory (DFT, [6]) works. Also, given that some of the results we present here rely on insights from stochastic representation decision theory (SRDT, [7]), we will give a short account of it.

**Recap of decision field theory.** Consider the simplest possible setup, a binary choice between two binary lotteries:

$$L_1 = \{o_A, p; o_B, \ 1-p\}, \qquad L_2 = \{o_C, \ q; o_D, \ 1-q\} \tag{1}$$

Lottery $L_1$ gives outcome $o_A$ (resp. $o_B$ with probability $p$ (resp. $1-p$) and $L_2$ analogously. In its simplest formulation (see [6] for the complete treatment), DFT assumes that the preference state $P$ of the DM at time $t$ is given by:

$$\begin{cases} P(0) = z \\ P(t) = z + \sum_{k=1}^t (V_2(k) - V_1(k)) \end{cases} \tag{2}$$

where $z$ denotes an anchor point (e.g. previous experience), while the time dependent subjective evaluations of alternatives read:

$$\begin{aligned} V_1(k) &= \omega_k(p)u(o_A) + \omega_k(1-p)u(o_B) \\ V_2(k) &= \omega_k(q)u(o_C) + \omega_k(1-q)u(o_D) \end{aligned} \tag{3}$$

In Eq (3), we have denoted with $u(\cdot)$ the DM utility function (assumed to be time independent) and with $\omega_k(p)$ a *random* realization at time $k$ of the attention weight devoted to the branch of lottery $L_1$ yielding outcome $o_A$. The formula captures the possibility that, from sample to sample, the DM may focus on different aspects of the lotteries. The evidence accumulation process continues until $|P|$ exceeds an inhibitory threshold $\theta$. Specifically, the probability of choosing

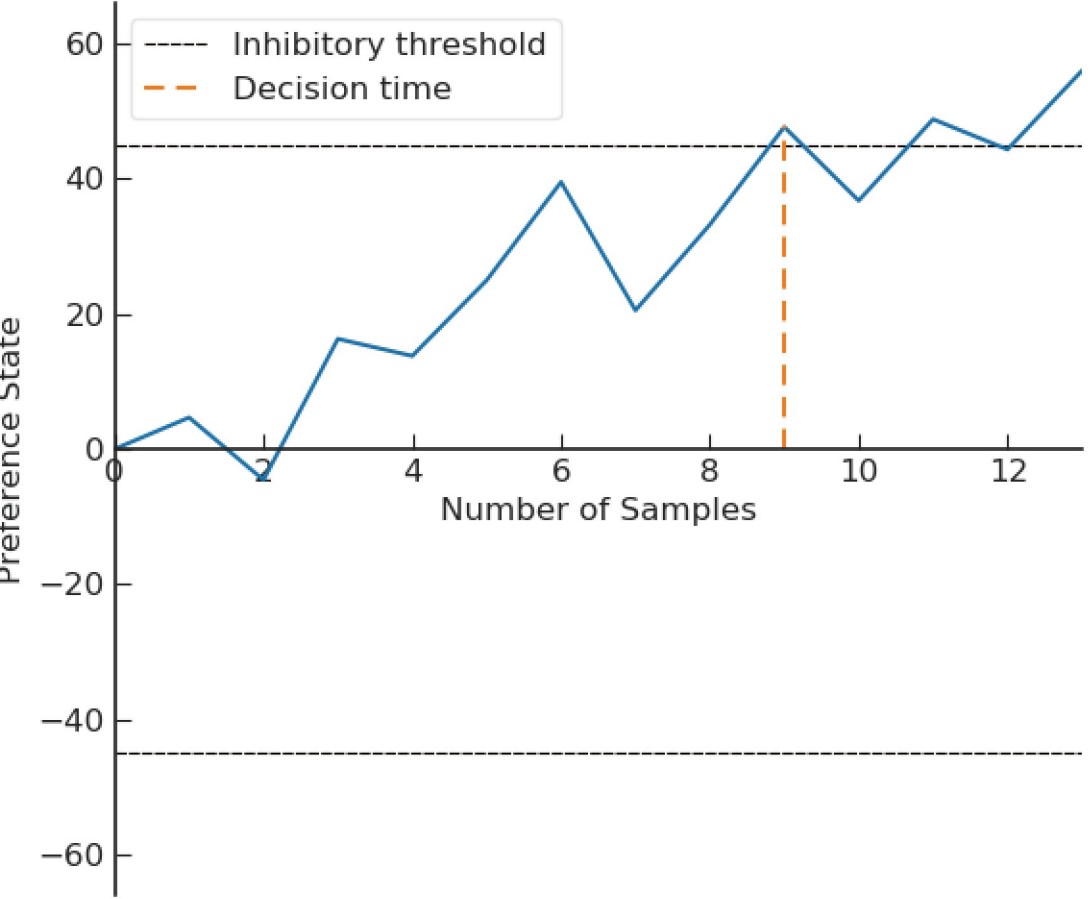

**Fig 1. Sample trajectory of preference state in DFT.** If the process reaches the upper boundary (resp. lower boundary) first, then lottery $L_1$ (resp. $L_2$) is chosen. Reproduced from [6].

lottery $L_1$ is given by:

$$\Pr(L_1) = Probability(P(t) > \theta \ \ \text{before} \ \ P(t) < -\theta) \tag{4}$$

A sample trajectory of the preference state $P$ is shown in Fig 1. At the empirical level, the theory is operationalized by four parameters. The valence difference $d = E[V_1 - V_2]$, its variance $\sigma^2 = Var[V_1 - V_2]$, the initial anchor point $z$, and the threshold criterion $\theta$.

**Recap of stochastic representation decision theory.** Similarly to DFT, SRDT describes the decision activity via a random walk, but with one key difference: outcomes and probabilities are combined in a non-symmetric and non-separable fashion as dual features of an event, rather than being simply multiplied to generate an index of worth. The idea is thus to capture the empirically observed "interaction" between subjective probability and subjective value [27–32]. People, for instance, tend to overestimate the likelihood of an event if the related consequence is negative.

For the decision task in Eq (1), the representative process takes the form of a random walk starting at the center of a "starfish" graph (see Fig 2). As before, we use random walk and its continuous version (Brownian motion) interchangeably. Each leg of the starfish has an absorbing boundary, and represents a probability-outcome pair of a lottery. The branch length encode information about the probability (higher probability corresponds to shorter branch),

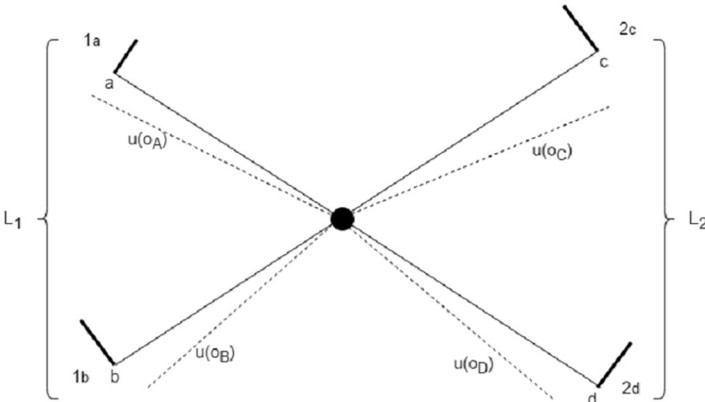

**Fig 2. Stochastic representation of the decision process between lotteries $L_1 = \{o_A, p; o_B, 1 - p\}$ and $L_2 = \{o_C, q; o_D, 1 - q\}$.** Branches $1_{a-b}$ (resp. $2_{c-d}$) represent the outcomes of $L_1$ (resp. $L_2$) and their related probabilities. The difference between the continuous and dashed lines represent the energy potential associated with the constant forces $\{u(o_A), u(o_B), u(o_C), u(o_D)\}$ exerted on the Brownian particle along each segment $\{1_a, 1_b, 2_c, 2_d\}$ respectively. The thick bars at the end of each branch depict the absorption boundary conditions. The segment lengths $\{a, b, c, d\}$ are determined by the objective probabilities $\{p, 1 - p, q, 1 - q\}$. The probability of choosing lottery $L_1$ (resp. $L_2$) is given by the probability of being absorbed along branch $1_a$ or $1_b$ (resp. $2_c$ or $2_d$).

while the potential energy along the leg encodes information about the outcome. Specifically, the Fokker-plank equation for the walker along, say, the branch representing outcome-pair $(o_A, p)$—denoted as $1_a$—reads:

$$
\begin{cases}
\dfrac{\partial p(x, t)}{\partial t} = u(o_A) \dfrac{\partial p(x, t)}{\partial x} + \dfrac{D}{2} \dfrac{\partial^2 p(x, t)}{\partial x^2} \\
p(a, t) = 0 \ \forall t \ (absorbing \ boundary) \\
p(0, t) = f(t) \ (probability \ from \ other \ branches)
\end{cases}
\tag{5}
$$

where $p(x, t)$ is probability density of finding the particle at position x (along branch $1_a$) and time t, $u(o_A)$ is the constant drift encoding the utility of outcome $o_A$ and $D$ is the diffusion coefficient embodying the noise level of the decision processing. The second line of Eq (5) enforces the absorbing condition at the end of the branch, while the third line accounts for the probability mass coming from other branches.

The probability $P_T(L_1, L_2)$ of choosing $L_1$ over $L_2$ at some time $t \leq T$ is then defined as the probability for the random walk to be absorbed along the branches pertaining to lottery $L_1$, conditioned on being absorbed somewhere before time $T$:

$$
\begin{aligned}
P_T(L_1, L_2) &= P(1_a | T) + P(1_b | T) \\
P_T(L_2, L_1) &= P(2_c | T) + P(2_d | T)
\end{aligned}
\tag{6}
$$

While there are conceptual similarities between DFT and SRDT, there are also clear differences. The first concerns the topology of the space where the random walker lives. In DFT, for any binary choice, the structure of the space is always simply one-dimensional, while in SRDT it depends on the number of outcome-probability pairs pertaining to each lottery. Second, the role of time-pressure is exemplified by the boundary distance (threshold $\theta$) in DFT, while SRDT makes use of a conditional absorption probability.

In terms of relative performance, SRDT is able to account for the role of time pressure on choice without an ad-hoc anchor point $z$. On the other hand, DFT can accommodate

violations of independence from irrelevant alternatives [33], while the "Luce" form [34] of the choice probability in SRDT rules it out.

Notwithstanding the structural differences, both models (as every other accumulator models) are essentially Markov processes. As argued in [35], Markov processes pose strong behavioural boundaries, regardless of its detail. For these models, choice acts simply as a way to *reveal* preference [36], i.e. the (choice) measurement by the experimenter does not affect the state of the decision-maker. However, extensive evidence (e.g. [19]) points to a constructive role of choice, i.e. people seem to select an optimization criterion that depends on the faced task. The constructionist view of choice bears a striking resemblance to the "scissors" analogy of Simon's paper [37]. According to Simon, human decision-making can be represented by a scissor in which one blade represents the cognitive capabilities of the decision-maker, while the other blade represents the situational context within which the decision-maker must choose. He imagined rationality as being shaped simultaneously by the environment structure and the human computational limits. The active role of measurement in quantum theory, which we now turn to in the sequel, may therefore serve as a powerful framework for describing preference construction.

## Decision theories based on static quantum models

**Brief review of the basics of quantum mechanics for decision theories.** Before reviewing the decision theories, we provide a brief primer on the mathematics of quantum mechanics, which will be used below.

*Hilbert space*. The operations of quantum mechanics take place in a $d$-dimensional Hilbert space $\mathcal{H}$, which is the span of $d$ orthonormal elements $\mathcal{H} = \text{span}\{|s_i\rangle, \ i = 1, \ 2, \ \ldots, \ d\}$. Each element of a basis $|s_i\rangle$ is written with the Dirac notation. The inner product of two elements satisfies $\langle s_i | s_j \rangle = \delta_{ij}$, where $\delta_{ij}$ is the Kronecker delta such that $\delta_{ij} = 1$ if $i = j$ and $\delta_{ij} = 0$ if $i \neq j$ and $\langle s_i |$ is the conjugate transpose of $|s_i\rangle$. Thereby, a Hilbert space is complete. Pure quantum states are represented by vectors in the Hilbert space.

*Quantum state*. A quantum state can be pure or mixed. A pure state is a unit vector $\psi\rangle$ in the Hilbert space and can be expressed as a linear combination of the elements of a basis:

$$|\psi\rangle = \sum_{i=1}^{d} a_i |s_i\rangle \tag{7}$$

where $a_i$ are the (complex) amplitudes of vector $|\psi\rangle$ projected on $|s_i\rangle$ and satisfy the normalization condition such that $|\psi|^2 = \langle \psi | \psi \rangle = \sum_{i=1}^{d} |a_i|^2 = 1$. Normally, a state $|\psi\rangle$ characterizes a decision-maker's mind when s/he is confronted with a decision task.

A mixed state, which describes an ensemble of pure quantum states or a quantum system entangled with other quantum systems via a partial trace, is written as a density matrix $\hat{\rho}$, and can be expressed as:

$$\hat{\rho} = \sum_i \omega_i |\psi_i\rangle \langle \psi_i| \tag{8}$$

where $\omega_j$ is the probability of state $|\psi_j\rangle$. If $\omega_i = 1$ and $\omega_{j \neq i} = 0$, the density matrix $\hat{\rho}$ describes a pure state. If $\hat{\rho} = I/d$, where $I$ is the identity matrix and $d$ is the dimension of the Hilbert space, the state is maximally mixed and it refers to the maximum *statistical* uncertainty or the minimum knowledge about the quantum system.

*Evolution of state.* A quantum state in a closed system evolves by applying a unitary opera-
tor $U(\tau)$, such that:

$$|\psi(t+\tau)\rangle = U(\tau)|\psi(t)\rangle \tag{9}$$

where $U(\tau) = \exp(-iH\tau)$. $H$ is a Hamiltonian operator, which is Hermitian ($H$ is equal to its
conjugate transpose) and time-independent in the case of a unitary operator. The evolution of
a density matrix by the same unitary operator is:

$$\hat{\rho}(t+\tau) = U(\tau)\hat{\rho}U^{\dagger}(\tau) \tag{10}$$

where $U^{\dagger}$ is a conjugate transpose of $U$. The unitary evolution of a quantum state can be analo-
gous to the change of mind during a decision process without perturbations. The non-unitary
evolution of a quantum state is associated with an open system (see Eq (15)).

*Measurement and observable.* A physical quantity of an entity that is in the form of a quan-
tum state can only be observed by taking appropriate measurement. An observable $A$ corre-
sponds to a physical quantity of interest. The operation to get the observable is called
measurement. In this paper, we are interested in projective measurements [38]. Thereby, the
observable $A$ is Hermitian and can be decomposed as follows:

$$A = \sum_i b_i \Pi_i^A \tag{11}$$

where $\Pi_i^A$ are the projection operators onto a basis of measurement and $b_i$ are corresponding
eigenvalues. Upon measuring observable $A$, the probability of getting a result $b_i$ from a pure
state $|\psi\rangle$ or a mixed state $\hat{\rho}$ is:

$$p(b_i) = \langle\psi|\Pi_i^A|\psi\rangle \quad \text{(pure state)} \quad \text{or} \quad p(b_i) = Tr(\Pi_i^A\hat{\rho}) \quad \text{(mixed state)}, \tag{12}$$

where $Tr(\cdot)$ is the trace of a matrix. The expected value of $A$ is:

$$\langle A\rangle = \langle\psi|A|\psi\rangle \quad \text{(pure state)} \quad \text{or} \quad \langle A\rangle = Tr(A\hat{\rho}) \quad \text{(mixed state)}, \tag{13}$$

After the measurement, the pure state $|\psi\rangle$ or the mixed state $\hat{\rho}$ *collapses* (or is updated) to a
new state corresponding to the result $b_i$:

$$|\psi\rangle \rightarrow \frac{\Pi_i^A|\psi\rangle}{|\Pi_i^A|\psi\rangle|^2} \qquad \text{(pure state)} \tag{14}$$

or

$$\hat{\rho} \rightarrow \frac{\Pi_i^A\hat{\rho}\Pi_i^A}{Tr(\Pi_i^A\hat{\rho})} \qquad \text{(mixed state)}. \tag{15}$$

This is the collapse postulate in quantum mechanics [39], distinguishing quantum behavior
to classical counterparts. In a decision theory based on quantum models, measurement often
refers to a decision action. The values elicited from observables determine the decision results.

Let us stress that the collapse postulate is indeed an additional *assumption* of quantum
mechanics, which does not necessarily apply to human decision-making. We thus need to be
careful in "importing" concepts from physics to the realm of choice theory. For example,
Yukalov et al. [40] use the quantum formalism only in the generalization of the classical Kol-
mogorov probability theory to the more general Hilbert space mathematics, leaving out the
collapse postulate. In contrast, Kvam et al. [41] make explicit use of the measurement

assumption in Eq (14) to model the effect of prior choice (i.e. a measurement) to a subsequent confidence rating.

**The essential ingredient: Quantum interference.** Quantum Probability Theory (QPT) may be seen as a trade-off between the strict axioms of Classical Probability Theory (CPT) and the loose structure of heuristic decision-making [42]. The QPT axioms are general enough to account for a wide host of behavioral biases, while preserving minimum rationality requirements, such as Dutch book consistency [43].

The main difference between the classical and quantum techniques is the way of calculating the probability of events. As soon as one accepts the quantum way of defining the probability, it generally becomes nonadditive and one immediately meets such quantum effects as interference and entanglement. Let us briefly outline the main ingredients of quantum interference. In CPT, given two events $E_1$ and $E_2$, the following condition always holds (law of total probability):

$$P(E_1) = P(E_1 \mid E_2)P(E_2) + P(E_1 \mid \bar{E}_2)P(\bar{E}_2) = P(E_1 \cap E_2) + P(E_1 \cap \bar{E}_2) \tag{16}$$

$P(E)$ denotes the probability of event $E$ and $\bar{E}$ is the complement of event $E$. In QPT, a state vector $|\psi\rangle$. (Eq (7)) lying in a complex vector space represents the system of interest (state of mind of the decision-maker). The probability of event $E$ (e.g. taking a particular decision) is given by (see Eq (12))

$$P(E) = |\Pi_E \psi\rangle|^2 \tag{17}$$

where $\Pi_E$ is the so-called projection operator and $|z| = \sqrt{Re(z)^2 + Im(z)^2}$ is the norm of complex number $z$. In words, to obtain the probability of an event, we first project the state vector onto the relevant subspace, and then take its squared length. The latter non-linear operation makes quantum probabilities generally non-additive. To see this, let us write the equivalent of Eq (16) in the quantum case:

$$
\begin{aligned}
P(E_1) &= |\Pi_{E_1}|\psi\rangle|^2 \\
&= |\Pi_{E_1}\Pi_{E_2}|\psi\rangle + \Pi_{E_1}(I - \Pi_{E_2})|\psi\rangle|^2 \\
&= |\Pi_{E_1}\Pi_{E_2}|\psi\rangle + \Pi_{E_1}\Pi_{\bar{E}_2}|\psi\rangle|^2 \\
&= |\Pi_{E_1}\Pi_{E_2}|\psi\rangle|^2 + |\Pi_{E_1}\Pi_{\bar{E}_2}|\psi\rangle|^2 + q \\
&= P(E_2 \cap E_1) + P(\bar{E}_2 \cap E_1) + q
\end{aligned}
\tag{18}
$$

In Eq (18), $I$ and $q$ denote the identity operator (identity matrix) and the quantum interference term. As anticipated, due to the non-linearity, there is an additional contribution to the probability of event $E_1$, encapsulated by the interference term $q = \langle \psi | \left( \Pi_{\bar{E}_2}^\dagger \Pi_{E_1} \Pi_{E_2} + \Pi_{E_2}^\dagger \Pi_{E_1} \Pi_{\bar{E}_2} \right) | \psi \rangle$, which violates the classical law of total probability. Let us also stress that in quantum mechanics $P(E_1 \cap E_2) \neq P(E_2 \cap E_1)$ in general, as the projection operators may not commute (matrix multiplication does not satisfy the commutative property). Many theories of quantum cognition have exploited the interference phenomena to explain several observed human "irrational" patterns, ranging from Probability and similarity judgments [44–46] to decision-making and Memory recognition [47, 48]. To exemplify, in

QDT, the probability of choosing a prospect (lottery) $\pi_n$ among $N$ alternatives is derived to be

$$p(\pi_n) = f(\pi_n) + q(\pi_n)$$
$$\sum_n p(\pi_n) = 1, \;\; 0 \leq p(\pi_n) \leq 1 \tag{19}$$

where the term $f(\pi_n)$ plays the role of classical probability, and the term $q(\pi_n)$ is the interference term. The detailed derivation of Eq (19) and the relevant explanation are provided in "S1 Appendix". As their classical counterparts, static quantum theories of decision-making are fundamentally unable to answer important questions, such as the effect of time pressure on choice and the distribution of response times [8, 9].

### Decision theory and quantum walks

**Motivation.** The most straightforward way to take advantage of both evidence accumulation models and quantum models is to implement quantum walks instead of random walks as the evidence accumulator [49]. The quantum walk can be implemented in quantum computing and outperforms classical computation in some tasks due to its quadratic or exponential speed up [50, 51]. The quadratic speed up is demonstrated by comparing the spread of probability distribution of a classical random walk and a quantum walk at the same evolution time (see Fig 3), where the variance of a classical walk scales as $t$ and that of a quantum walk scales

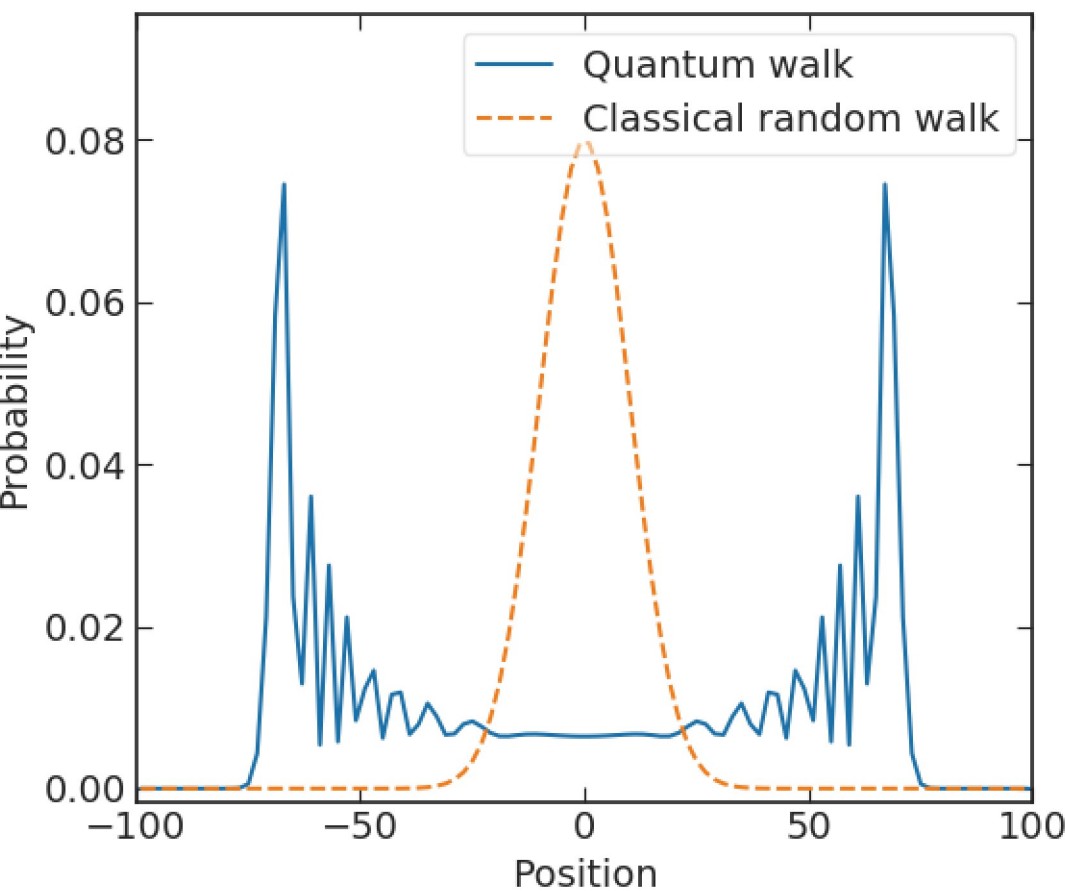

**Fig 3. The probability distribution of a discrete-time quantum walk (blue line) and a classical random walk (orange dash line).**

as $t^2$ [23]. Another dimension of benefits of quantum walks is that, with appropriate topological structures, computation based on quantum walks retrieves a universal quantum computer [52].

**Previous decision theories based on quantum walks.** In sub-section "Stochastic decision theories: drift diffusion model, DFT, SRDT", we reviewed several evidence accumulation models that capture the dynamical process of decision-makings. In sub-section "Decision theories based on static quantum models", we reviewed the essential quantum mechanical tools and discussed their usefulness in cognition science and economics. Static quantum models provide a strong explanatory tool, but they ignore dynamical properties of choice. For example, a decision theory based on a static quantum model hardly encompasses response time of a decision. We hence focus on a class of quantum dynamical models, namely quantum walks, which conceptually takes advantages of both evidence accumulation and quantum tools.

To explain human choice, there are two possible formulations of quantum walks: continuous- and discrete-time quantum walk, henceforth denoted as CTQW and DTQW. Both walks are implemented on discrete space (We omit the walks on continuous space due to the following reasons: i) there is yet to be a discrete-time quantum walk on a continuous space; ii) the characteristic propagation (e.g. ballistic spread) of continuous-time quantum walk cannot arise in continuous space [53]). Busemeyer et al. [54] developed the first decision theory based on continuous-time quantum walks. Their quantum walk model outperforms the classical random walk models in predicting the dynamics of a task consisting of decision and confidence rating [55]. Fuss and Navarro [10] used a discrete-time quantum walk to reconstruct a cooperative and competitive parallel (CCP) model that better accounts for excitatory and inhibitory effect of neural process. Both walks are typically implemented in one-dimensional space to depict a binary-choice task (DFT employs a similar structure, see Fig 1). As in a Markov random walk or a drift-diffusion process, the average position of the quantum walker shifts towards one end of the space, which corresponds to a drift effect. In general, a binary-choice decision is revealed by measuring the position of the walker. Nevertheless, the meaning of the distribution and the process of making a decision is very different from a Markov random walk or a drift-diffusion process. The differences also project to a different understanding of the decision-making process [35].

In spite of the advantages of a decision model based on quantum walks, several studies indicate that a mixed walk, which is an open quantum walk interacting with an external environment, better captures the decision process [10, 35, 56]. In the physical world, a pure quantum state is hardly isolated. Thus, decoherence of the state occurs due to energy dissipation or any other interactions with the environment. Decoherence is responsible for the transition from quantum to classical probabilities [39]. By analogy, Busemeyer et al. [57] used a function that combines a quantum walk and a Markov random walk by distributing weight on each. Fuss and Navarro [10] implemented a constant decoherence method. In sub-section "Decoherence" we will discuss different decoherence procedures, each having a particular behavioral implication.

Let us now focus on the limitations of previous theories. This will serve as a motivation for our new framework presented in Section "Framework of decision theory based on discrete-time quantum walks". Busemeyer et al. [54] proposed a CTQW to simulate a drift-diffusion process governed by the Schrödinger equation instead of the Fokker–Planck equation. The model formulation is similar to DFT, where a decision is made if a particle in a 1-D bounded space wanders beyond the threshold towards either of the boundaries. As for DFT, the wandering of the particle represents evidence accumulation. The preference is represented by the drift that biases the walk. However, when a drifting force is introduced, space discretization introduces the effect of localization in CTQW, whereas a similar effect is absent in a drift-diffusion

process. The reason is as follows. In discrete space, the potential differs from site to site by a finite amount $\Delta V$, leading the "scattering" of the quantum wave function and multiple interferences. This can lead to a phenomenon called Anderson localization [58, 59], corresponding to the spatial trapping (localization) of the walk by the multiple interferences. The significance of the localization effect depends on the value of $\Delta V$. Regarding the decision theory based on CTQW, such an effect does not capture any cognitive process as far as we know.

In contrast, DTQW simulates a drifting effect with a different machinery (see sub-section "Interference effect of choice on confidence") and the localization phenomenon is absent. Moreover, an extra degree of freedom of DTQW, namely the spin state, provides more possible ways to bias the walk, which accounts for different endogenous and exogenous factors that influence decision-making (see sub-section "Influence of parameters on the probability distribution" for effects of parameters and Section "Examples of application of the proposed quantum walk framework to decision-making" for examples that employ these effects in decision theories). Moreover, the previous decision theory based on DTQW [10] implements only a specific type of walk, called the Hadamard walk. As there are many other quantum walks, restricting to the Hadamard walk limits the breadth of possible representations of decision theories.

To sum up, with a generalized DTQW on the one hand, we: i) avoid the localization effect of CTQW; ii) go beyond the Hadamard walk, showing that our framework has much richer characteristics to develop better decision theories. To gain further insight on decision theories based on both types of walks, we first review a model based on CTQW in this section. As already mentioned, regarding DTQW, the theory developed by Fuss and Navarro [10] relies on a special case, namely the Hadamard walk, of our general framework (see Section "Framework of decision theory based on discrete-time quantum walks").

*The continuous-time quantum walk model by Busemeyer et al.* A continuous-time quantum walk is constructed similarly to a Markov random walk. For a simple Markov walk, the probability of a particle to be present at position $x \in Z$ at a given time $t$ is written as $p(x, t)$. The evolution of the local probability depends on the preceding neighboring probabilities:

$$p(x, \ t) = qp(x - 1, \ t - 1) + (1 - q)p(x + 1, \ t - 1) \tag{20}$$

The time-dependent distribution is a vector $\mathbf{p}(t) = (\ldots, p(x - 1, t), p(x, t), p(x + 1, t), \ldots)^T$. The evolution of the local probability at $(x, t)$ gives a global evolution $\mathbf{p}(t + 1) = \mathbf{Q}.\mathbf{p}(t)$ where $Q$ is the the so-called transition matrix whose entry $Q_{ij}$ is the probability of transitioning from position i to position j in one time step. From expression (20), we have then $Q_{i-1,i} = 1 - q$, $Q_{i+1,i} = q$ $\forall i \in Z$.

For a quantum walk, the common interpretation is that a particle lives in a superposition of states at each given time, assuming a definite position upon performing a measurement. In the decision theory proposed by Busemeyer et al. [54] based on CTQW, if the length of a finite 1-D space is $L$, the positions space $x \in [0, L]$ is discretized as a set of individual positions $\{|x_i\rangle, i = 0, 1, \ldots, N\}$. The separation between successive positions is $\Delta = L/N$. The Hilbert space is given by the span of $\{|x_i\rangle\}$. If we take the continuum limit $\Delta \to 0$, the Hilbert space has infinite dimension and the position space become continuous. However, the continuous-space quantum walk has distinct characteristics and is out of the scope of this paper. The state $|\psi(t)\rangle$ of the particle representing the decision process can be written as a linear combination of positions at time $t$:

$$|\psi(t)\rangle = \sum_i a_i(t)|x_i\rangle \tag{21}$$

where $a_i(t)$ is the amplitude of the state at position $x_i$ at time $t$. The probability distribution at time $t$ in the position space is given by $p(x_i, t) = |a_i(t)|^2$. The time evolution of the particle state reads $|\psi(t)\rangle = U\psi(t_0)\rangle$ with $U = \exp(-iH(t - t_0))$. $H$ is a $N \times N$ matrix with the following entries:

$$H_{ii} = -\frac{\mu_i}{\Delta}, \qquad H_{i-1, \ i} = H_{i+1, \ i} = -\frac{\sigma^2}{\Delta^2} \qquad (22)$$

where $\mu_i$ is the potential at position $x_i$, accounting for drifting forces and $\sigma$ is the diffusion parameter that determines how fast the probability distribution spreads. All other entries of $H$ are zero. As in other types of quantum walks, the variance of the probability distribution spreads as $t^2$.

The probability distribution in the position space of a quantum walker represents a decision mind analogously to an evidence accumulation model, where the position of the classical random walk in a 1-D space represents a confidence level or a preference towards a prospect. Nevertheless, there is a remarkable difference in the representation of choice or confidence level. The probability distribution can be monitored at each time step if a Markov random walk is implemented. In contrast, the distribution is not accessible unless a measurement is performed, which corresponds to the operator $M = \sum_i |x_i\rangle\langle x_i|$ being applied to the system state. As mentioned in sub-section "Decision theories based on static quantum models", this modifies the state $\psi\rangle$, disturbing the evolution of the walk. From theories based on evidence accumulation models, the distribution is supposed to be constantly monitored by a decision-maker without being disturbed. The perturbative nature of observations or measurements during the evolution of a quantum system creates large differences between these two types of theories.

Kvam et al. [55] used a continuous time quantum walk as an evidence accumulator to represent a two-step decision task, with the first step being a decision and the second step a confidence rating on the chosen hypothesis. Because of the active role of measurements in quantum mechanics, they successfully predict the disturbance of the confidence assessment by the presence of prior decision and the oscillatory confidence level with decision time. Both patterns cannot be easily accounted for by a Markov random walk. We present a similar application of a discrete-time quantum walk in sub-section "Interference effect of choice on confidence".

## Framework of decision theory based on discrete-time quantum walks

### Mathematical basics

According to the reasoning in previous sections about the advantages of evidence accumulation models and quantum static models in describing human decision behavior, we construct a general framework for decision theories based on DTQW. Through the Feynman formulation of quantum mechanics [38], the model represents the decision as the result of an array of parallel accumulator models that interact in both constructive and destructive ways. This mirrors schematically the existence of many units of information treatment with coexistence of excitatory and inhibitory couplings in the brain.

As already mentioned, a first DTQW model of decision was developed by Fuss and Navarro [10], using the Hadamard quantum walk. However, this model fit does not significantly outperform a classical model. Here, we study a more general walk, where more degrees of freedom are introduced, including the initial spin distribution and the two parameters of the quantum coin operator.

**Hilbert space.**  Compared to CTQW, it is not possible to construct a non-trivial DTQW if the state of the particle is characterized only by its position [23]; an extra degree of freedom,

spin, is added. Thus, the walk takes place on the Hilbert space $\mathcal{H} = \mathcal{H}_s \otimes \mathcal{H}_p$, where $\mathcal{H}_s \equiv Span\{|\leftarrow\rangle, |\rightarrow\rangle\}$ is a two-dimensional spin space and $\mathcal{H}_p \equiv Span\{|x\rangle, \ x \in \mathbb{Z}\}$ is a position space with infinite dimension. The basis $|\leftarrow\rangle, |\rightarrow\rangle$ of the two-dimensional spin are annotated with left and right signs as they are associated with left- and right-moving evolution of the state.

**State of the particle.** The state of the particle is generally time dependent and can be written as:

$$|\psi(t)\rangle = \sum_{x \in \mathbb{Z}} a(x, t) \begin{pmatrix} L(x, \ t) \\ R(x, \ t) \end{pmatrix} \otimes |x\rangle = \sum_{x \in \mathbb{Z}} \begin{pmatrix} \psi_L(x, \ t) \\ \psi_R(x, \ t) \end{pmatrix} \otimes |x\rangle \qquad (23)$$

The state is a linear combination of Kronecker products of spins and positions. We can describe it as the wave packet of a particle, analogous to that in the physical world. At each position $x$, there is a spin state $(L(x, t), R(x, t))^T$ coupled to a position $|x\rangle$. The spin state describes the spin distribution at each position and is a unit vector so that $|L(x, t)|^2 + |R(x, t)|^2 = 1$. Here the letters $L$ and $R$ represent the two components of the spin state, namely left spin and right spin. The spin degree of freedom is crucial in determining the left- or right-moving evolution in the position space (see Eq (28) below). The amplitudes $a(x, t)$ are complex numbers and indicate the probability distribution in the position space. They satisfy the normalization condition $\sum_{x \in \mathbb{Z}} |a(x, t)|^2 = 1 \ \forall t \geq 0$. It is convenient to write the amplitude and spin state collectively as the characteristic state $(\psi_L(x, t), \psi_R(x, t))^T$ at position $x$ and time $t$. Usually, in a decision theory, the state of the particle represents the status of the decision-maker's mind, which changes with time during the decision process.

Additionally, The formalism using density matrices is the most general to describe a quantum system. As introduced in sub-section "Decision theories based on static quantum models" with expression Eq (8), a density matrix can be written as $\hat{\rho}(t) = \sum_i \omega_i |\psi_i(t)\rangle\langle\psi_i(t)|$ where $\omega_i$ is the probability of the $i$-th possible pure state $\psi_i(t)$.

**Evolution.** The state evolves by applying an unitary operator $U$ so that $|\psi(t)\rangle = U^t|\psi(0)\rangle$. The state of the quantum particle is redistributed over the position space by a shift operator followed by a quantum coin operator. We define the following coin operator

$$C = \begin{bmatrix} e^{i\xi}\sqrt{\rho} & e^{i\zeta}\sqrt{1-\rho} \\ e^{-i\zeta}\sqrt{1-\rho} & -e^{-i\xi}\sqrt{\rho} \end{bmatrix}, \qquad (24)$$

shift operator

$$S = |\leftarrow\rangle\langle\leftarrow| \otimes \sum_{x \in \mathbb{Z}} |x-1\rangle\langle x| + |\rightarrow\rangle\langle\rightarrow| \otimes \sum_{x \in \mathbb{Z}} |x+1\rangle\langle x|, \qquad (25)$$

and total unitary operator

$$U = S \cdot (C \otimes I) \qquad (26)$$

In the simplest case, the quantum coin operator $C$ is uniform across the position space. If the coin operators depend on the position $x$, we can write the coin operators as $C_x$ instead. Parameters $\xi$ and $\zeta$ control the bias of the walk, while $\rho$ is related to its variance [60] (not to be confused with $\hat{\rho}$, the density matrix operator). The shift operator moves the particle one step to the left (resp. to the right) if its spin is left (resp. right). If $\xi = \zeta = 0, \rho = 0.5$, the coin is a Hadamard matrix and the quantum walk becomes a Hadamard walk. By applying $U$ to $|\psi(t)\rangle$ for $\tau$

time steps, we have:

$$|\psi(t+\tau)\rangle = U^\tau|\psi(t)\rangle \tag{27}$$

The coin operator acts on the spin state at each position $x$, and the shift operator move the wave packet at position $x$ towards $x-1$ or $x+1$. As already reported for the CTQW, the variance of the position distribution $\sigma^2$ scales as $t^2$ (ballistic spread), in contrast with the classical diffusive behavior $\sigma^2 \sim t$ [49]. Similarly to Eq (20), the dynamics of the quantum walk in position space can be described by the following system of two coupled difference equations:

$$\begin{pmatrix} \psi_L(x,\ t) \\ \psi_R(x,\ t) \end{pmatrix} = \begin{pmatrix} e^{i\xi}\sqrt{\rho}\psi_L(x+1,\ t-1) + e^{i\zeta}\sqrt{1-\rho}\psi_R(x+1,\ t-1) \\ e^{-i\zeta}\sqrt{1-\rho}\psi_L(x-1,\ t-1) - e^{-i\xi}\sqrt{\rho}\psi_R(x-1,\ t-1) \end{pmatrix} \tag{28}$$

From Eq (25), the shift operator splits the state at position $x$ into left- and right-moving components. If $\xi = \zeta = 0$ and $\rho = 1$, the left- and right-moving components at position $x$ and time $t$ becomes $\psi_L(x, t)$ and $-\psi_R(x, t)$, respectively, which means that the distribution on position space propagates without deformation and independently along the left and right directions. If $\xi = \zeta = 0$ and $\rho = 0$, the left- and right-moving components invert the dependence on $\psi_L$ and $\psi_R$. In sub-section "Influence of parameters on the probability distribution", we further analyze the effects of parameters quantitatively.

From Eq (27), the corresponding evolution for the density matrix of the state is:

$$\hat{\rho}(t+\tau) = U^\tau\hat{\rho}(t)(U^\dagger)^\tau \tag{29}$$

where $U^\dagger$ is the conjugate transpose matrix of $U$.

**Initial state.** The initial state $|\psi(0)\rangle$ normally represents the state of mind prior to the decision stimulus. It is usually assumed symmetric around some initial position $x_0$. For instance, we can prepare a state where the spin state is independent of position and write it as

$$|\psi(0)\rangle = \begin{pmatrix} L_0 \\ R_0 \end{pmatrix} \otimes \sum_{x\in\mathbb{Z}} a(x,0)|x\rangle \tag{30}$$

The spin state will in general entangle with position state after evolution, so that $|\psi(t)\rangle$ cannot be written as a product state (i.e. tensor product between a vector living in $\mathcal{H}_s$ and a vector living in $\mathcal{H}_p$). The simplest initial state is $|\psi(0)\rangle = (L_0, R_0)^T \otimes |0\rangle$. An analytic study of time-dependent solutions of the quantum state $|\psi(t)\rangle$ with this initial state in an unbounded space is performed in sub-section S2.1 of "S2 Appendix", in which we follow the same procedure of [23], but use a general coin operator instead of the Hadamard operator. Given that a closed form solution is not obtainable, we study it mainly through numerical simulations.

**Measurements.** Two types of basic measurement can be performed. If we take the measurement $M_x = I_s \otimes |x\rangle\langle x|$ at position $x$, where $I_s$ is the identity matrix in the spin space, we obtain the probability of finding the particle at $x$ at time $t$ as:

$$p(x) = \langle M_x\rangle_\psi = \langle\psi(t)|M_x|\psi(t)\rangle = |a(x,t)|^2 \tag{31}$$

We can also measure the global spin state with

$$M_s = |s\rangle\langle s| \otimes I_x \tag{32}$$

where $|s\rangle \in \{|\leftarrow\rangle, |\rightarrow\rangle\}$, to obtain the corresponding probabilities for the global spin to be $|\leftarrow\rangle$

or $|\rightarrow\rangle$ respectively:

$$p(L) = \langle M_L \rangle_\psi = \langle\, \psi(t)|M_L|\psi(t)\rangle = \sum_{x=-\infty}^{+\infty} |L(x,t)|^2 \tag{33}$$

$$p(R) = \langle M_R \rangle_\psi = \langle\, \psi(t)|M_R|\psi(t)\rangle = \sum_{x=-\infty}^{+\infty} |R(x,t)|^2 \tag{34}$$

By performing measurements, the particle state transforms into a new state, as described in Eq (14). A pure state therefore becomes an ensemble of several pure states, or a mixed state. The map representing the collapse of quantum states can be described by an operation-sum representation [61]:

$$\epsilon(\hat{\rho}(t)) = \sum_i M_i \hat{\rho}(t) M_i^\dagger \tag{35}$$

where $M_i$ belong to a complete set of measurements that satisfy $\sum_i M_i^\dagger M_i = I$ and $I$ is the identity matrix. The expression in Eq (35) is based on a positive operator-valued measure (POVM) [61]. In the case of projective measurements in this paper, $M_i$ are orthogonal projectors. Hence, the POVM reduces to a projection-valued measure (PVM).

**Boundaries of the walk.** The walk can be implemented on an unbounded space or bounded space. There are two types of boundaries that can be inserted to either one or both sides of the 1-D space, namely reflecting and absorbing boundaries. The simplest way to add an absorbing boundary at position $\bar{x}$ is to set the coin parameter $\rho = 1$ at $x \geq \bar{x}$; the reflecting boundary requires only $\rho = 0$ at position $\bar{x}$.

The intuition is the following. If the wave packet enters an absorbing boundary at $\bar{x}$ from $\bar{x} - 1$ for the first time at time $t$, from Eq (28) we know that $\psi_L(\bar{x}, t) = 0$ and only $\psi_R(\bar{x}, t)$ has a non-zero magnitude, since the wave packet cannot be in $\bar{x} + 1$, implying $\psi_L(\bar{x} + 1, t) = \psi_R(\bar{x} + 1, t) = 0$. The evolution at the next step transmits $\psi_R(\bar{x}, t)$ to the right such that $\psi_R(\bar{x} + 1, t + 1) = -e^{-i\xi} \psi_R(\bar{x}, t)$ according to Eq (28). Since $\psi_L(\bar{x}, t) = 0$, there is no left-moving component shifting to position $\bar{x} - 1$ at time $t + 1$. Now, for all positions $x > \bar{x}$, $\rho = 1$. Hence, the evolution will continue passing the $\psi_R$ to the right. As $\psi_L$ is always zero for $x > \bar{x}$, there is no left-moving components for all the wave packet to the right of the boundary. As a result, the wave packet entering $\bar{x}$ at time $t$ will keep traveling to the right without returning, the same for the wave packets entering $\bar{x}$ at subsequent time steps.

The probability that the particle gets absorbed by the boundary can be found by taking the measurement from Eq (31) at the locations beyond the absorbing boundary. In contrast to a classical random walk, the wave packet cannot be completely absorbed due to a striking property of quantum walks. This phenomenon was extensively studied for example by Meyer [62]. Qualitatively, the part of the wavepacket that is transmitted at the boundary reduces the interference between the reflected wave packet and the incoming wave packet one step before the boundary. This reduces overall destructive interference, and creates an effective reflection at the boundary. The effect of incomplete absorption is very distinct from a classical counterpart. From [23], for a quantum walker on a line where only one absorbing boundary is present, there is a finite probability for the walker to escape to infinity and not be absorbed by the absorbing boundary. In the classical case, any absorbing boundary at a finite distance from the starting point will eventually absorb the particle. An even more bizarre phenomenon occurs if we add a second absorbing boundary on the other side of the walk. In the classical case, the presence of the second absorbing boundary reduces the absorbing probability of the first one. In the quantum case, placing a second absorbing boundary increases the absorbing probability

at the first boundary. This is because the second absorbing boundary reflects a small proportion of wave packets, such that eventually more wave packets arrive at the first boundary.

An equivalent way to set an absorbing boundary is through taking the measurement Eq (31) at the boundary at each time step. In this case, the evolution is better represented by a density matrix since the measurement causes decoherence and results in a mixed state. The evolution operation followed by a measurement at each time step maps a state as:

$$\hat{\rho}(t+1) = \sum_i M_i U \hat{\rho}(t) U^\dagger M_i^\dagger \tag{36}$$

where $M_i$ is a set of position measurements at boundaries at $x = b_1, b_2$ and positions other than boundaries $x \neq b_1, b_2$, which can be written as:

$$M_{1,2} = \sum_{x=b_1, \ b_2} I_s \otimes |x\rangle\langle x|, \qquad M_3 = \sum_{x \neq b_1, \ b_2} I_s \otimes |x\rangle\langle x| \tag{37}$$

A reflecting boundary is represented by the coin operator with $\rho = 0$ at the boundary. If a wave packet enters the boundary at $x$ for the first time from the left at time $t$, we have $\psi_L(x, t) = 0$ and $\psi_R(x, t)$ is non-zero, as before. The next evolution at position $x$ sends $\psi_R(x, t)$ to position $x - 1$, so that $\psi_L(x - 1, t + 1) = e^{i\zeta}\psi_R(x, t)$. Then it sends $\psi_L(x, t)$ to position $x+ 1$, which is zero. This means that the operator at $x$ inverts the right-moving component to a left-moving component, which embodies the reflection.

**Observables.** For an unbounded space or a bounded space with reflection, the useful observables are simply the spin (left or right) or the position of the particle or the conjunction of the two. For a space bounded by absorbing boundaries, the information carried by spin and position will eventually get destroyed by the absorption. Therefore, knowing the outcome that the particle is absorbed by either of the boundaries is more useful. The most straightforward application of the framework is binary decision tasks. The bias of a distribution in the spin space or in the position space, or the absorption probability by two boundaries, are taken as proxies for choice preferences. The position of the particle can also represent the degree of belief or confidence.

## Influence of parameters on the probability distribution

There are many parameters controlling the evolution of a quantum walk. In this subsection, we focus on the ones that influence the variance in the position space and the bias in the position or the spin space. We formulate the variance $var_p$ and the biases $bias_p$ in the position space and $bias_s$ in Eq (38). By assigning a value 0 to the left spin and a value 1 to the right spin (the value assigned does not affect the result), we can define the following:

$$var_p = \sum_{x \in \mathbb{Z}} (x - \mu_p)^2 p(x)$$

$$bias_p = \frac{\sum_{x \in \mathbb{Z}} (x - \mu_p)^3 p(x)}{\left[\sum_{x \in \mathbb{Z}} (x - \mu_p)^2 p(x)\right]^{3/2}} \tag{38}$$

$$bias_s = \frac{(0 - \mu_s)^3 p(L) + (1 - \mu_s)^3 p(R)}{\left[(0 - \mu_s)^2 p(L) + (1 - \mu_s)^2 p(R)\right]^{3/2}}$$

where $var_p$ and $bias_p$ are the variance and bias (or skewness) of the distribution in the position

space, $bias_s$ is the bias of distribution in the spin space. $\mu_p$ and $\mu_s$ are the mean in position and in the spin space, which can be expressed as:

$$\mu_p = \sum_{x \in \mathbb{Z}} x p(x)$$

$$\mu_s = 0 \cdot p(L) + 1 \cdot p(R)$$

(39)

The variance of a quantum walk is a factor contributing to the decision time and decision outcome at different times. The bias of the distribution in the position space or spin space, depending on the representation we choose for decision-making, is a measure of the tendency of choice at different times.

To remove redundant parameters, we first degenerate the parameters using an analysis similar to that by Tregenna et al. [60]. Let the particle start at position $x = 0$. Consider the simple initial state:

$$|\psi(t=0)\rangle = \begin{bmatrix} L_0 \\ R_0 \end{bmatrix} \otimes |0\rangle = \begin{bmatrix} e^{i\alpha}\sqrt{\eta} \\ e^{i\beta}\sqrt{1-\eta} \end{bmatrix} \otimes |0\rangle$$

(40)

where $\alpha$, $\beta$ and $\eta$ are such that $|L_0|^2 + |R_0|^2 = 1$. Consider also the most general Coin operator in Eq (24), with its parameters $\rho$, $\xi$, $\zeta$. In sub-section S2.2 of "S2 Appendix", we show that, without loss of generality, we can constrain several parameters and still obtain all the possible evolutions of the walk. For simplicity, we use the initial state $|\psi(t=0)\rangle = \left(\sqrt{\eta}, \; i\sqrt{1-\eta}\right)^T \otimes |x_0\rangle$ where we fix $\alpha = 0$, $\beta = \pi/2$ and $x_0$ is the initial position of the particle. As for the coin, we can set $\zeta = 0$ and only vary $\xi$ and $\rho$. Therefore, we are left with three parameters: $\rho$, $\xi$ in the coin operator and $\eta$ in the initial state. If we generalize the initial state to Eq (30), the parameters act similarly on the properties of the walk, since the initial spin distribution is invariant across the position space.

We exemplify the influence of parameters in Fig 4 and summarize the results in Table 1. The following more detailed analysis is based on the plots in sub-section S2.3 of "S2 Appendix". According to the plots, a larger $\rho$ increases the variance when the walk is symmetric or biased by $\xi$. This is similar to the effect of evolution time on the distribution in the position space. However, when the walk is biased by $\eta$, increasing $\rho$ first widens the spread, then the variance drops as $\rho$ approaches 1. This is because $\rho$ enhances the bias due to $\eta$, and in the extreme case where $\rho = 1$, there is only a single peak on one side, resulting in a minimum variance. We also observe that a larger $\rho$ does not change the direction of the bias in the position space, but enhances the bias due to $\eta$ and eliminates the bias due to $\xi$.

According to Plot iv in Fig 4, the variation of $\xi$ mainly governs the bias of the probability distribution in both spaces and has less effect on the variance. The value of $\xi$ ranged from $-\pi$ to $\pi$. An oscillatory pattern with a period $2\pi$ is observed, where the bias in both spaces is zero at $\xi = 0$, peaks at $\xi = -\pi/2$ and reaches the minimum at $\xi = \pi/2$. Hence, we drop out the repetitive domain of $\xi$ and narrow it down to $-\pi/2 < \xi < \pi/2$ in the following.

The parameter $\eta$ also affects the bias, with a minor influence on the variance. If $0 < \eta < 0.5$, the walk is biased towards the left, whereas $0.5 < \eta < 1$ leads to a right-biasing walk. $\eta = 0.5$ corresponds to a symmetric walk.

Observed from all the plots, the bias in the spin space positively correlates with the bias in the position space. From the plots ii and iii, varying parameter $\rho$ has an oscillatory effect on the bias in the spin space. This effect is similar to the oscillatory behavior of spin states over time, which will be presented in sub-section "Interference effect of choice on confidence". The

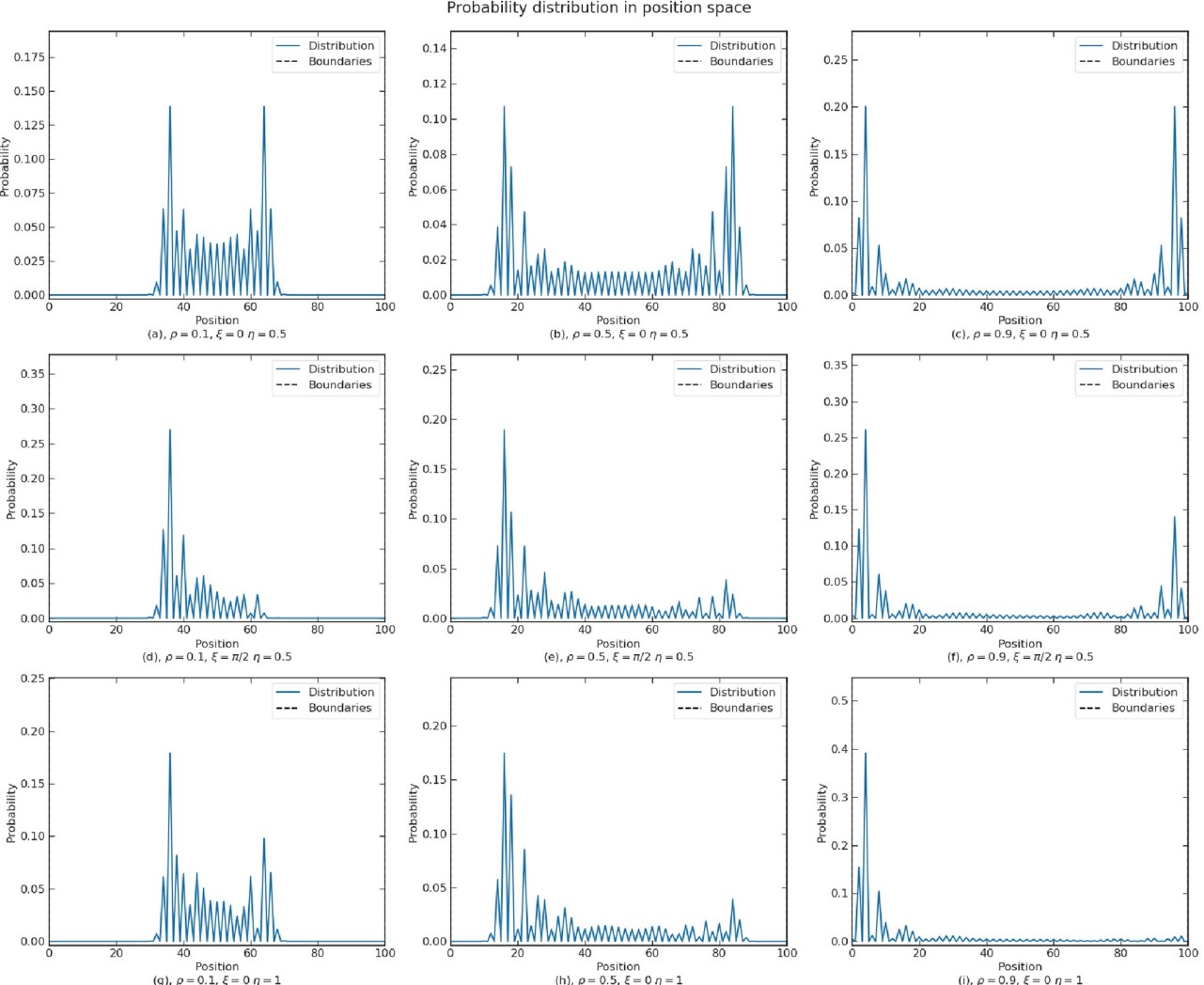

**Fig 4. Influence of parameters $\rho$, $\xi$ and $\eta$ on the probability distribution of a walk in the position space at 50 time steps with no boundaries at both sides.**

**Table 1. Summary of the influence of parameters on a walk.**

| Influence of parameters on the walk | | | |
|---|---|---|---|
| **Characteristics of the walk** | $\rho \in [0, 1]$ | $\xi \in \left[-\frac{\pi}{2}, \frac{\pi}{2}\right]$ | $\eta \in [0, 1]$ |
| $var_p$ | $var_p$ increases with $\rho$ when the walk is unbiased or biased by $\xi$; it rises and drops with increasing $\rho$ if the walk is biased by $\eta$ | $var_p$ changes periodically with increasing $\xi$ | $var_p$ rises and drops with increasing $\eta$ |
| $bias_p$ | $bias_p$ is not effected if the walk is not biased by other parameters; it rises and drops with increasing $\rho$ if the walk is biased by $\xi$; it is amplified by $\rho$ if the walk is biased by $\eta$ | The walk is biased to the right with $\xi \in \left[-\frac{\pi}{2}, 0\right]$; biased to the left with $\xi \in \left[0, \frac{\pi}{2}\right]$ | The walk is biased to the right with $\eta \in [0, 0.5]$; biased to the left with $\eta \in [0.5, 1]$ |
| $bias_s$ | Similar to the above, with fluctuations | Same as the above | Similar to the above, with fluctuations |
| Others | A changing $\rho$ in the position space produces a continuous drifting effect | | |

oscillatory effects caused by both $\rho$ and time confirm the similarity of the impacts of the two parameters. Plot ii shows that the bias in the two spaces are weakly correlated.

Parameters $\eta$ and $\xi$ determine the bias encoded in the initial state before the evolution starts. Parameter $\eta$ determines the spin distribution of the initial state. From the study in [63], parameter $\xi$ of the coin operator only affects the bias at the position at which the particle is initialized. In fact, parameter $\xi$ only affects the bias at the first time step, as documented by extensive numerical simulations. However, the preference can still change after exposition to new evidences in some decision scenarios. We thus need a method that biases the walk during the evolution. In sub-section "Interference effect of choice on confidence", we introduce a model that allows a continuous drifting effect by gradually changing the parameter $\rho$ of the coin operator across the position space. If $\rho$ is smaller on the left side and larger on the right side of the position space, the particle drifts to the right, and vice versa.

## Decoherence

A pure quantum state can lose its coherence and become a mixed state. In a decision process, this might occur due to the interaction between a decision mind and some environment that distracts or disturbs the decision process. A quantum walk with absorbing boundaries experiences a constant decoherence, as the particle is continuously probed by performing measurements that implement the effective absorption process. If a decision-maker "checks" the position of the particle from time to time to evaluate her confidence in making a choice (a sort of introspection), this also leads to decoherence. From [56, 57], the decoherence effect is modeled as a linear combination of a quantum walk and a classical random walk with corresponding weights. This mixture may be difficult to justify as a representation of the deliberation process of a decision-maker. Fuss and Navarro [10] introduced decoherence through a wave amplitude damping model, commonly employed in quantum computing. With this method, the spin distribution is completely shaped by the environment under full decoherence. Here, we introduce another method to generate decoherence for a pure quantum state.

Suppose that at each time step there is a chance $p_m$ that a position and/or spin measurement is performed on the quantum walk. The probability $p_m$ that a measurement occurs is then a measure of the level of decoherence. If $p_m = 0$, no measurement is performed and the state remains pure. If $p_m = 1$, the state undergoes maximum decoherence. Sets of measurement based on the position basis, the spin basis, and both bases are defined as $\{M_p\}$, $\{M_s\}$, $\{M_{p,s}\}$:

$$
\begin{aligned}
\{M_p\} =\ & \{\sqrt{p_m} \cdot I_s \otimes |x\rangle\langle x| : x \in \mathbb{Z},\ \sqrt{1-p_m} \cdot I_s \otimes I_p\} \\[4pt]
\{M_s\} =\ & \{\sqrt{p_m} \cdot |\leftarrow\rangle\langle \leftarrow \otimes I_p,\ \sqrt{p_m} \cdot |\rightarrow\rangle\langle \rightarrow | \otimes I_p,\ \sqrt{1-p_m} \cdot I_s \otimes I_p\} \\[4pt]
\{M_{p,s}\} =\ & \{\sqrt{p_m} \cdot |\leftarrow\rangle\langle \leftarrow | \otimes x\rangle\langle x| : x \in \mathbb{Z},\ \sqrt{p_m} \cdot |\rightarrow\rangle\langle \rightarrow | \otimes |x\rangle\langle x| : x \in \mathbb{Z}, \\[4pt]
& \sqrt{1-p_m} \cdot I_s \otimes I_p\}
\end{aligned}
\tag{41}
$$

The evolution is determined by Eq (36). The probability distributions associated with measurement on different bases and different levels of decoherence $p_m$ are plotted in Fig 5. The maximum decoherence level $p_m = 1$ always retrieves the distribution of a classical random walk in all plots, which coincides with the findings by Aharonov et al. [49]. This is expected theoretically since a measurement at each time steps amounts to full decoherence where all interference effects are suppressed. The classical random is recovered as a result of the probabilistic nature of the quantum measurement process, which is akin to tossing a random coin at each measurement steps. While the evolution of the quantum walk is purely deterministic, it is the measurement process that introduces stochasticity.

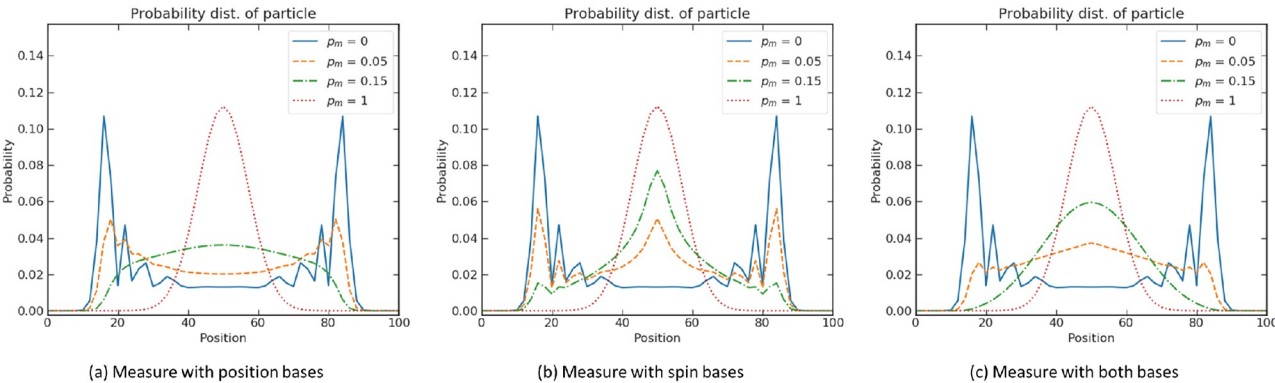

**Fig 5. Position probability distribution with different levels of decoherence $p_m$.** (a) Measurement in position space. (b) Measurement in spin space. (c) Measurement in both spaces. Parameters are $\rho = 0.5$, $\eta = 0.5$, $\xi = 0$, step sizes $s_L = s_R = 1$ at time $t = 50$ in an unbounded position space.

Fig 5 shows that a small value of $p_m$ (0.05, 0.15) already gives a characteristic shape of a random walk (the distributions become more concentrated at the centre), which is very different from the quantum walk where $p_m = 0$. As the quantum walk spreads with quadratic speed compared to the classical random walk, the variance of the distribution in the position space acts as a measure of the quantumness.

Fig 6 shows the variance at time $t = 50$ as a function of the level of decoherence $p_m$. One can observe that the variance drops fast as $p_m$ increases from 0 for all types of measurement. The rates with which the variance decreases for increasing $p_m$ are slightly different for the three types of measurement, with the measurement based on both bases leading to the fastest decrease. This quantifies the degree of vulnerability of the quantum walk in keeping its quantum characteristic in the presence of measurements.

The level of decoherence or quantumness can be interpreted in the context of decision theories as follows. As described by Yukalov and Sornette [64], quantum interference captures deviations from rational behavior. Decoherence is responsible for the disappearance of interference terms, retrieving classical behavior. Our formulation of decoherence in terms of probabilistic measurements can be descriptively interpreted in the following way: if a decision-maker constantly "checks" her/his state of mind, the behavior will shift towards a non-interference mode. This resonates with the evidence that continuous probing and self-assessment make the decision-maker more rational. A specific prediction of decoherence in terms of relationship between computational time and probabilistic distortions, anticipating a speed-accuracy tradeoff, is presented in sub-section "Probability Judgment".

## Examples of application of the proposed quantum walk framework to decision-making

### Probability judgment

**Perception of probability modeled by absorption probability.** A question often arises in cognition science and economics: how do we perceive the probability or the risk of the occurrence of an event? The perception of probability is often referred to as subjective probability. There are different techniques to assess subjective probability (see the review [65]). Typically, people exhibit an inverse S-shaped pattern of the probability weighting function (e.g. red line in Fig 7a) [66, 67]. In essence, subjects tend to overestimate small probabilities and underestimate large probabilities. The empirical result from another study [68] produced a consistent pattern of the mapping from objective to subjective probabilities. At the same time, people

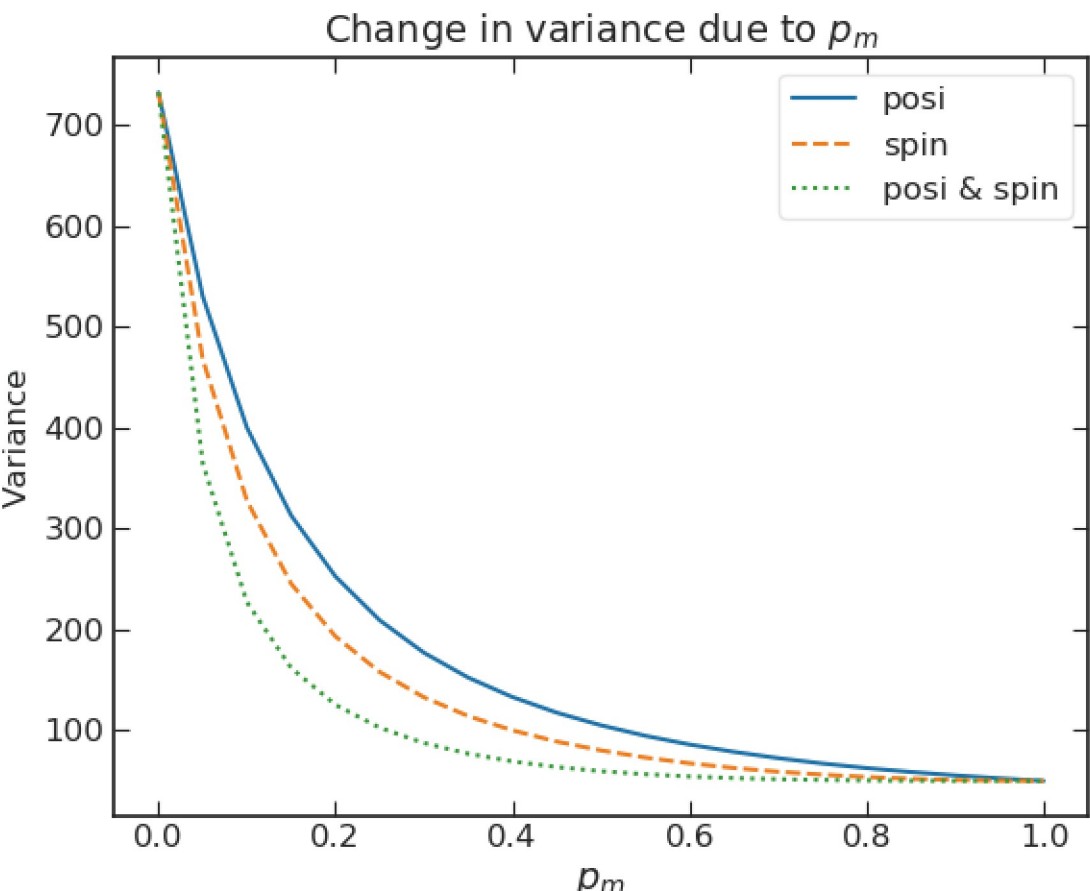

**Fig 6. Variance of the mixed walk vs $p_m$ for the three types of measurement discussed in the text with parameters $\rho = 0.5$, $\eta = 0.5$, $\xi = 0$, step sizes $s_L = s_R = 1$ at time $t = 50$ in an unbounded position space.**

tend to "ignore" very small probability events and consider as certain very high probability ones [69]. The two patterns together result in an inverse double-S-shaped curve (e.g. green line in Fig 7a). As there is no clear inversion point of probability estimation between "small probability" and "probability", the model we propose in this section is able to produce a flexible inversion point.

The idea is to represent the process of probability understanding by a random walk, wandering in some abstract space whose properties encode objective probabilistic information. Suppose a particle moves on a segment stochastically with identical small steps; each step brings the particle randomly to either left or right with equal chances. If the distance from the starting point and one end (say, left end) is $a$, and the distance from the other end is $b$, given infinite time, it is a standard result in classical diffusion theory that the probability that a *classical* particle hits the left end first is $b/(a + b)$. Inspired by DFT and SRDT reviewed in subsection "Stochastic decision theories: drift diffusion model, DFT, SRDT", we contruct an evidence accumulation model of probability perception. Since the hitting or absorbing probability exactly reflects the ratio of distances for a classical random walk, it is interesting to see how this mapping changes if a quantum walk is used instead. A model developed from the DTQW framework with degenerate parameters and a simple initial state has been described in sub-section "Influence of parameters on the probability distribution", where the coin operator and the

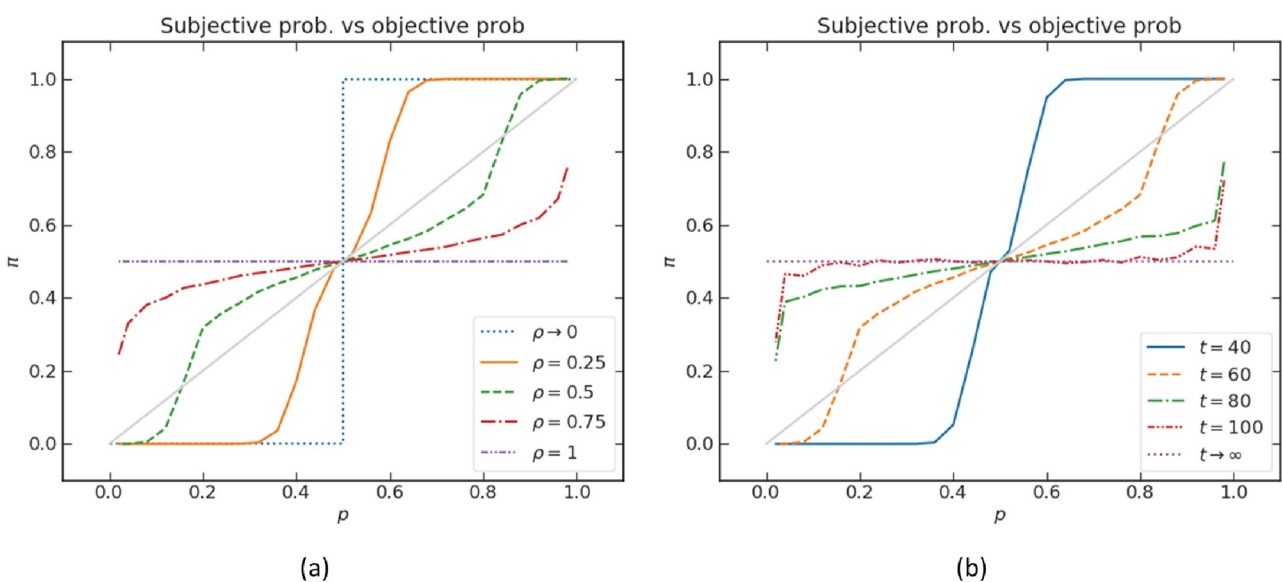

**Fig 7. Subjective probability $\pi$ vs objective probability $p$.** The objective probability $p$ is encoded in the relative lengths of branches explored by the quantum walk, as explained in the text. The walks have unbiased parameters $\xi = 0$, $\eta = 0.5$. The distance between the two boundaries is kept as 50. In (a), $\pi(p)$ is plotted for different values of $\rho$ at fixed $t = 60$; in (b), $\pi(p)$ is plotted for different values of $t$ at fixed $\rho = 0.5$.

initial state are:

$$C = \begin{bmatrix} e^{i\xi}\sqrt{\rho} & \sqrt{1-\rho} \\ \sqrt{1-\rho} & -e^{-i\xi}\sqrt{\rho} \end{bmatrix}, \qquad |\psi(0)\rangle = \begin{bmatrix} \sqrt{\eta} \\ i\sqrt{1-\eta} \end{bmatrix} \otimes |x_0\rangle \tag{42}$$

The particle starts at position $x_0$ ($x_0 > 0$). There are two absorbing boundaries at $x = 0$ and $x = N$. If the "understanding" of probability is modeled through a classical random walk, given enough time, a subject's valuation of the probability of an event is taken as being objective. With a classical random walk, as we have mentioned, the probability of being absorbed summed over all times at one end of the interval is equal to the ratio of the distance from the initial position to the other end divided by the total distance between the two ends of the interval. Therefore, we encode the objective probability $p$ for the choice encoded in the left end as the ratio of the distance between the starting point and the right end to the total distance:

$$p = \frac{N - x_0}{N} \tag{43}$$

Our object of interest is the probability that the particle is absorbed by the left boundary at time $t$, *given* that an absorption event occurs. We denote this conditional absorption probability as $\pi(t)$. Its expression reads

$$\pi(t) = \frac{p_L(t)}{p_L(t) + p_R(t)} \tag{44}$$

where $p_L(t)$ is the absorption probability at $x = 0$ and $p_R(t)$ is the absorption probability at $x = N$, at time $t$. Eq (44) can be understood in the frequentist interpretation of probability: by launching a large number (ideally infinite) number of particles, $\pi(t)$ is computed as the ratio between the number of left absorptions to the total absorption events.

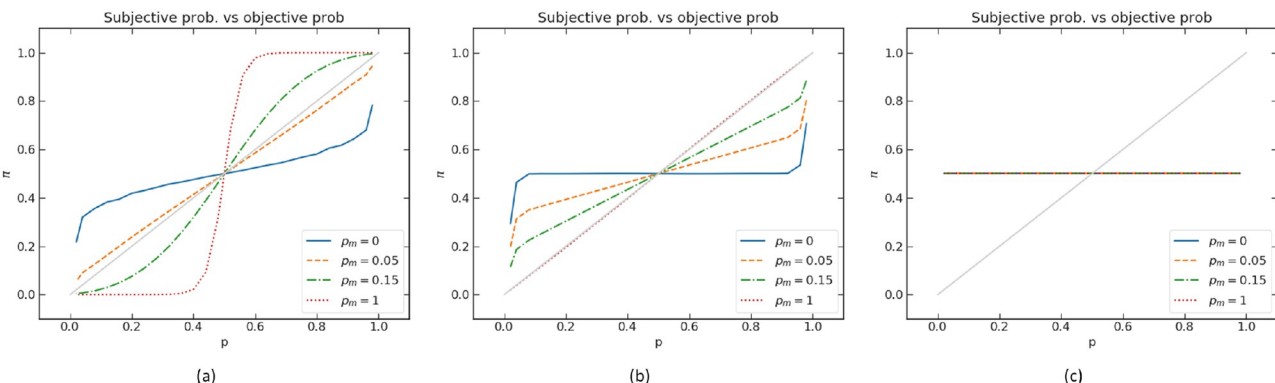

**Fig 8. Subjective probability $\pi$ vs objective probability $p$ for different values of the probability $p_m$ that a position and/or spin measurement is performed on the quantum walk.** The objective probability $p$ is encoded in the relative lengths of branches explored by the quantum walk, as explained in the text. The parameter of the walks are (a) $t = 75$, $\rho = 0.5$. (b) $t \to \infty$, $\rho = 0.5$. (c) $t = 75$, $\rho = 1$.

We take $\pi(t)$ as a proxy for representing the subjective perception of probability by human subjects. Given a topology (distance ratio between absorbing walls) encoding the objective probability of an event $p$, its perception $\pi(t)$ depends on the underlying stochastic process and the available time for deliberation. In the case of a classical random walk and infinite deliberation time, $\pi(t) \xrightarrow[t \to \infty]{} p$ as already mentioned. Let us now study what happens if the underlying stochastic process is a quantum walk.

With the representation of deliberation by a quantum walk, the evaluation of the objective probability becomes a tradeoff between efficiency and accuracy. A fully coherent quantum walk, with its ballistic spread, quickly reaches a probability estimate (corresponding to a smaller mean absorption time), which however is quite far from the objective probability (Figs 7 and 8). Instead, when the walk experiences decoherence, via successive measurements, the stochastic process becomes slower (more like a diffusive spread), but the probability estimate is more accurate (Fig 8b). Behaviorally, we can interpret the successive measurements as a "conscious check" of the decision-maker to constantly be aware and in control of its probability evaluation process. This operation is costly, resulting in longer processing time. In this sense, probabilistic distortions arise when a quick and dirty evaluation is needed, in agreement with evolutionary justifications of cognitive biases [70].

In our first analysis, we choose the unbiased parameter configuration $\xi = 0$, $\eta = 0.5$. The subjective probability $\pi(t)$ is plotted as a function of different objective probabilities $p$ in Fig 7, for different values of $\rho$ and $t$. The dashed black diagonal line indicates the equivalence map of $\pi(t) = p$, representing a perfect understanding of objective probability.

From Fig 7, at finite time and $0 < \rho < 1$, we observe inverse (double)-S-shaped curves for approximately $t > 60$ or $\rho > 0.5$. Some curves show an inverse double-S shape (e.g. at $t = 60$ or $\rho = 0.5$), describing the subjective perception of extremely small or large probabilities. A similar result was obtained from SRDT (Fig 9 in [7]), where the inverse double-S-shaped subjective probability arises through the interplay of value-distortion (probability of an event influenced by its magnitude) and time pressure. The model here does not need value distortion, but assumes that the underlying stochastic process is quantum in nature, thus with intrinsic interference between different positions along the deliberation paths.

The two analyzed parameters $\rho$ and $t$ play a similar role in regulating the subjective probability. They both widen the spread of the walk. Therefore, different values of $\rho$ and $t$ result in similar curves. Behaviorally, $\rho$ can be interpreted as regulating an "internal time constraint", for example to save computational resources, even when there is no explicit time pressure. The

parameter $t$ may be thought of as exogenous (e.g. a time limit imposed by the experimenter or the environment). In the special cases where $t \to \infty$ or $\rho = 1$, we have $\pi = \frac{1}{2}$ regardless of the value of the objective probability $p$ (purple dashed lines in Fig 7). While this appears nonsensical, one should note that a fully coherent quantum process hardly describes a real decision-maker, who is more adequately thought of as an open quantum system interacting with its surrounding environment. Indeed, for the asymptotic case $t \to \infty$, a much more reasonable probability estimate is obtained as soon as the system is subjected to *small* decoherence. This can be seen by studying the walk with different levels of decoherence $p_m$ (see also sub-section "Decoherence)". The choice of measurement bases for decoherence is not crucial as they have similar effect on the variance of the distribution in position space, here we choose the measurement on the position basis. Fig 8 shows the mixed walk with $t = 75$, $\rho = 0.5$, where one can observe that the patterns with different $p_m$ are very similar to those found in Fig 7. This is not particularly surprising because $p_m$ is also associated with how fast the distribution spreads. We plot the graph where $t \to \infty$, $\rho = 0.5$ and $t = 75$, $\rho = 1$ with different levels of decoherence in Fig 8b and 8c, respectively. It demonstrates that, at infinite time, the subjective probability $\pi(t)$ still depends on $p$ for $p_m > 0$. However, with $\rho = 1$, $\pi(t)$ does not depend on $p$ regardless of the value $p_m$. The quantum coin parameter $\rho$ thus controls the sensitivity of the probability judgment, independent of t and $p_m$.

This sensitivity may change under certain circumstances. For example, in order to save computational time, a decision-maker may "choose" to increase $\rho$ to speed up the spread of the distribution, which causes loss of accuracy. On the other hand, decoherence slows down the evaluation and makes it more accurate. The evaluation of the objective probability is essentially a tradeoff between efficiency and accuracy. In addition, the full decoherence retrieves a rational evaluation. Conceptually, this coincides with the result of QDT by Yukalov and Sornette [15], in which utility theory is recovered if no quantum interferences are present.

Apart from the inverse (double)-S-shaped curve, another prediction from this model is that the evaluation of small probabilities shifts from underestimation to overestimation with increasing available time. Correspondingly, the overestimation of large probability shifts to underestimation over time. Although in this section we focused on probability judgment rather than value-based choice, the shift of probability evaluation may refer to the inversion of risk attitude with increasing decision times. Clearly, to make actual predictions in this context, the subjective probability needs to be embedded in a choice theory (e.g. rank-dependent utility, [71]).

There are other possibilities to encode the objective probabilistic information, for example in the initial spin state. An alternative is reported in "S3 Appendix", showing qualitatively similar results.

**A conceptual explanation of the conjunction fallacy.** The conjunction fallacy described by Tversky and Kahneman [13] reveals a strikingly irrational behavior of humans in the context of probabilistic judgment. An event with specific conditions is considered more likely to happen than an event with a more general condition. In the famous Linda's example, given a short description of Linda, the majority of people judged less probable the event $A$ "Linda is a bank-teller" with respect to the event $A \bigwedge B$ "Linda is a bank-teller and is active in the feminist movement". It is clear that, from a rational perspective, this is absurd: indeed, in accordance to the axioms of classical probability theory, for any two events $A$ and $B$, we should have $P(A \wedge B) \leq P(A)$. Nevertheless, most people followed the reverse inequality.

The representation of subjective probability in terms of the absorption probability of a quantum walk suggested in the previous section allows for a qualitative account of the phenomenon. As reported by Ambainis et al. [23], considering a quantum Hadamard walk with

only one absorbing wall that is to the left of the starting position, the probability that the quantum walk exits to the left is $\frac{2}{\pi}$. If another absorbing boundary is placed at location $n$ to the right, the probability that the walk exits to the left actually *increases*, approaching $\frac{1}{\sqrt{2}} > \frac{2}{\pi}$ in the limit of large $n$, corresponding to a more than 10% increase. Such behavior is strikingly different from the classical case, where adding another absorbing boundary clearly decreases the probability of being absorbed by the first one. The origin of such behavior is quantum interference: the addition of a second absorbing boundary on the right removes a part of the quantum "waves", which would otherwise have interfered destructively with another part of the "waves" reaching the left boundary.

Consider now the following mapping:

- $P(A)$ is given by the probability for a quantum walk to exit to the left when only a left absorbing boundary is present. The left boundary represent the statement "Linda is a bank-teller".

- $P(A \wedge B)$ is given by the probability for a quantum walk to exit to the left when an additional boundary to the right is present. The right boundary accounts for the statement "Linda is active in the feminist movement", apparently reducing the number of walks terminating at the left boundary.

Upon decoherence (sub-section "Decoherence"), the quantum walk becomes classical and retrieves $P(A \wedge B) < P(A)$ for *any* bias and boundary locations. In contrast, depending on the specific bias (coin operator) and the location of the walls, the fully coherent quantum walk may reverse the inequality.

While the Conjunction fallacy has already been examined and explained in terms of quantum interference [45], the novelty here is that the representation of subjective probability, identified via the absorption probability of a one-dimensional quantum walk, explains both the conjunction fallacy and S-shaped probabilistic distortions, connecting two seemingly disparate phenomena. Because probability perception is described as a trade-off between efficiency and accuracy (see sub-section "Perception of probability modeled by absorption probability"), the model comes with a *qualitative* prediction: people making the conjunction fallacy respond faster than those who do not, which is in agreement with empirical findings [72].

## Interference effect of choice on confidence

As mentioned in sub-section "Motivation", Kvam et al. [55] used a continuous-time quantum walk as an evidence accumulator to represent a two-step decision task. The perceptual decision task they consider presents a set of randomly moving dots, with a certain proportion moving towards the left or right, on a screen. A prior decision task is for a subject to determine the net drifting direction of the set of dots. A confidence rating task requires the subject to rate how much s/he believes the dots to drift to the left or right, with a rating scale from 0 (surely left) to 100 (surely right). With respect to a setup where subjects are asked only the confidence rating, the prior decision causes either bolstering or suppression effect, depending on the elapsed time between the action of prior decision and confidence rating.

A fundamental assumption in [55] is that both tasks can be associated with measurements in the same space, i.e. the position space. Specifically, the probability of giving a confidence rate $x$ is represented by the probability of finding the walker at position $x$, while the probability of choosing left (resp. right) is the probability of finding the walker at some position $x \in [0, 50[$ (resp. $x \in ]50, 100]$). While this can be a reasonable first approximation, this procedure for discrete choices is somewhat artificial. A priori, the distinct nature of the task (discrete choice vs continuous confidence rating) points to a representation in two different spaces. By employing

a discrete-time quantum walk rather than the continuous one, the spin degree of freedom, due to its inherent discreteness, is a natural candidate to represent choice probability, while we leave the position degree of freedom to represent confidence rating. The correlation between the two, described in Section "Framework of decision theory based on discrete-time quantum walks", ensures that the mapping does not lead to inconsistencies.

To retrieve an analogy with the CTQW model, we prepare a DTQW model for a case of left-drifting dots as follows. A finite position space is prepared with $x \in [0, N]$. The coin operators, prepared according to

$$C_x = \begin{pmatrix} 0 & 1 \\ 1 & 0 \end{pmatrix}, \qquad x = 0, \ N \tag{45}$$

$$C_x = \begin{pmatrix} \sqrt{\rho} & \sqrt{1-\rho} \\ \sqrt{1-\rho} & -\sqrt{\rho} \end{pmatrix}, \qquad \rho = \frac{x}{N-1} \ \text{for} \ x \in [1, \ N-1] \tag{46}$$

are different across space. The coin operators at boundaries $x = 0, N$ cause reflecting effects. Increasing parameter $\rho$ of the coin operators from left to right results in a drifting-like effect towards the left boundary (supposing the net motion of the dots is toward the left). For this perceptual decision task, a symmetric initial distribution in position and spin spaces is assumed, modeling an unbiased view before the evidence accumulation has started. The initial state is set as follows

$$|\psi(0)\rangle = \frac{1}{\sqrt{2}} \begin{pmatrix} 1 \\ i \end{pmatrix} \otimes \sum_{x=0}^{N} \frac{1}{\sigma\sqrt{2\pi}} e^{-\frac{1}{2}\left(\frac{x-x_0}{\sigma}\right)^2} |x\rangle \tag{47}$$

where $x_0 = 50$ is the center of the position space, the value of $\sigma$ is 1/6 of the distance between the two boundaries, indicating a distribution of initial positions concentrated at the center of the position space. The initial state is the Kronecker product of an unbiased spin state and a position state distributed as a normal distribution across the position space.

Fig 9 shows that the wave packets are drifting towards the left boundary and keep bouncing back and forth near the reflecting boundary, which is analogous to the results of the CTQW model (Fig 5 in [54]). The confidence rating $x$ is equated to the probability of detecting the particle at position $x$, see Eq 31. If the particle is detected closer to the left boundary, the subject is more confident about the left-drifting choice, and vice versa. The probability of choosing left (resp. right) is identified with the probability of measuring the spin in the left state (resp. in the right state), see Eq (32).

With the above construction, a particle starts to evolve at $t = 0$. In the first case (choice and confidence rating), a measurement is performed on both the position basis and the spin basis, leading to decoherence. The evolution of the corresponding state can be described by Eq 36. In the second case (only confidence rating), the walk evolves without disturbance. Fig 10 compares the expected position of the walk, which represents the confidence rating, with and without the prior decision. The prior decision time is $t = 50$. We observe an obvious disturbance with the prior decision. In the parts of the graphs where the expected position of the disturbed walk is above that of the undisturbed walk, this corresponds to the bolstering effect (confidence rating is amplified); in the parts of the graphs where the expected position of the disturbed walk is below that of the undisturbed walk, this corresponds to the suppression effect (confidence rating is weakened). The observations comply with the studies of the CTQW model.

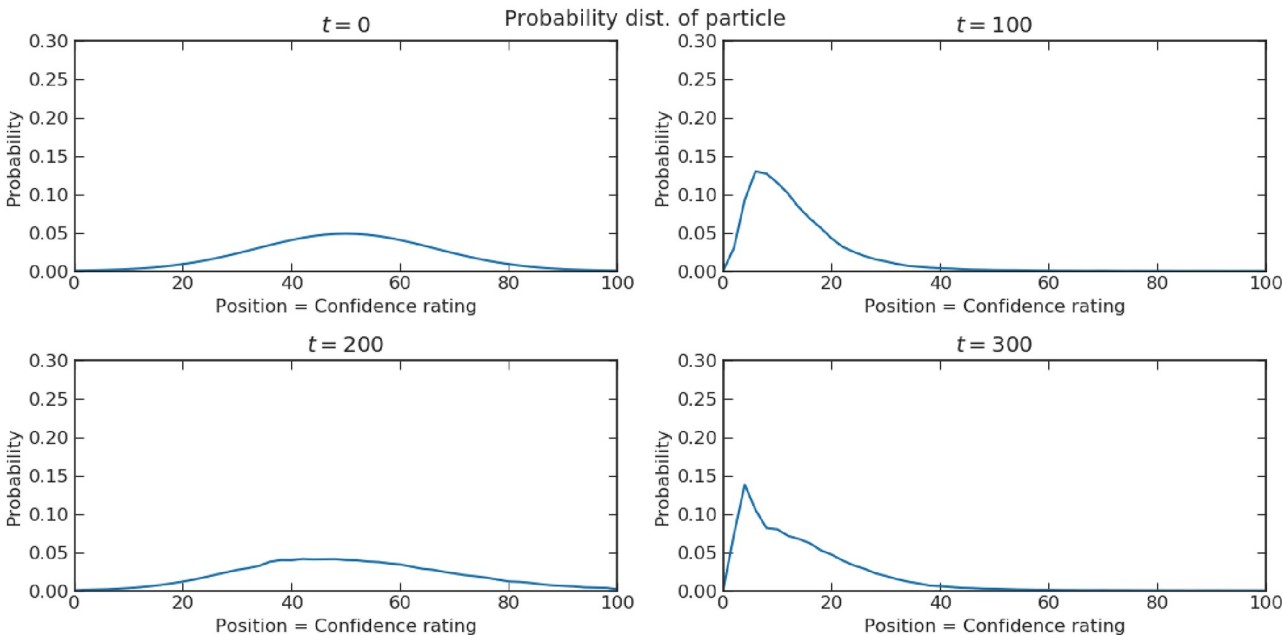

**Fig 9. Probability distribution of the discrete-time quantum walk (DTQW) described in the text in the position space at different times.**

## Conclusion

We have presented a short review of discrete-time quantum walks with the aim of fostering further research on their potential applications to model human decision-making. We suggest that discrete-time quantum walks (DTQW) may be more suitable than the continuous-time version (CTQW), in the goal of combining the strands of literature on evidence accumulator models and on the quantum formalism of cognition. Due to its additional spin degree of freedom, the DTQW allows for a natural modeling of model choice and confidence rating in separate bases (see sub-section "Interference effect of choice on confidence"). In the context of probabilistic judgment, we provided a toy evidence accumulation model of probability perception, which describes the understanding of probability as a trade-off between efficiency and accuracy. This simple model comes with interesting quantitative (S-shaped probability weighting) and qualitative (faster response when making the conjunction fallacy) predictions.

Our framework emphasizes the relationship between response times and type of preferences, which has been extensively studied across several domains (see [73] and references therein). Despite the mixed evidence, it is generally believed that faster responses are associated with heuristic thinking, while slower ones come from deliberative reasoning. In his book "Thinking, Fast and Slow", Kahneman [74] suggested that our decision-making results from the interaction of two conflicting "systems" of reasoning, System I (fast and instinctive) and System II (slow and deliberative). In this spirit, dual process theories have been shown to quantitatively explain many puzzling phenomena [75]. However, an on-going debate is present about their neurological support [76]. Here, at least conceptually, we have proposed an alternative interpretation of the relationship between response time and type of response. Referring to the probability judgment (sub-section "Probability Judgment"), there is only one process going on, which is perturbed and transformed over time by probabilistic measurements (self-assessments). The evolution of the process is controlled by the level of decoherence $p_m$. This has the effect of slowing down the walk, making it also more accurate. Within our CTQW framework, the two fast and slow systems of [74] are replaced by a single system, but with two

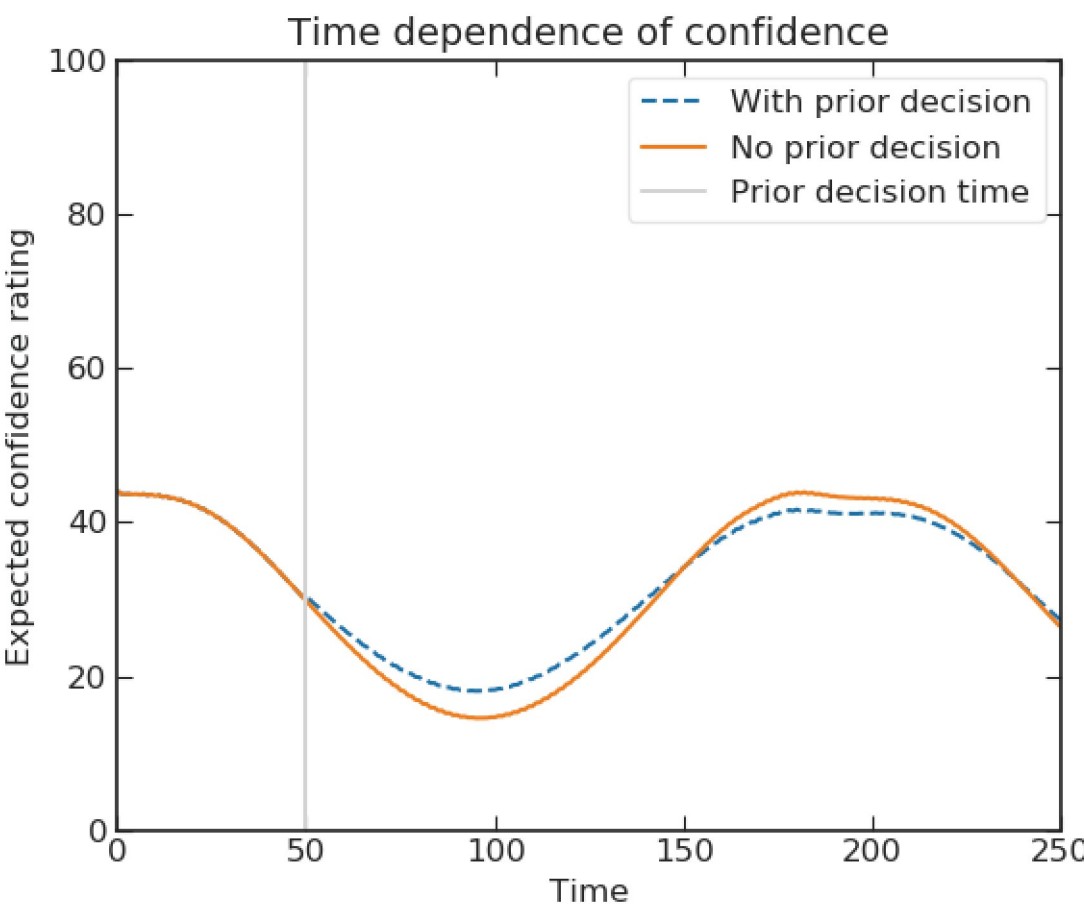

**Fig 10. Expected confidence rating vs time.** The orange curve shows the evolution of the expected position of the particle without a prior decision, the blue curve indicates the counterpart with a prior decision. The prior decision time is at $t = 50$. In the parts of the graphs where the expected position of the walk with prior decision is above that of the walk with no priori decision, there is a bolstering effect (confidence rating is amplified) and vice-versa.

types of self-assessment or introspection. The "thinking fast" regime is obtained with no or little self-assessment, while the "thinking slow" regime corresponds to a strong rate of self-assessment.

Of course, the structural properties which have been presented through simple examples are not sufficient to falsify the models. At this stage, the parsimony of its formulation and the wealth of obtained properties, which are in qualitative or semi-quantitative agreement with empirically observations, makes our framework interesting to further explore.

## Supporting information

**S1 Appendix. Mathematical foundation of quantum decision theory.**
(PDF)

**S2 Appendix. Mathematical properties of discrete-time quantum walks.**
(PDF)

**S3 Appendix. An alternative way of encoding objective probabilistic information.**
(PDF)

## Acknowledgments

We are grateful to Emma Schepers whose master thesis contributed to a first partial exploration of some of the themes developed in the present paper. The work was supported in part by the Shenzhen Science and Technology Innovation Commission (grant no. GJHZ20210705141805017).

## Author Contributions

**Supervision:** Didier Sornette.

**Writing – original draft:** Ming Chen, Giuseppe M. Ferro.

**Writing – review & editing:** Ming Chen, Giuseppe M. Ferro, Didier Sornette.

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
