## [Decision Letter · Decision Letter 0]

20 Jun 2022

PONE-D-22-13566On the use of discrete-time quantum walks in decision theory PLOS ONE

Dear Dr. Chen,

Thank you for submitting your manuscript to PLOS ONE. After careful consideration, we feel that it has merit but does not fully meet PLOS ONE’s publication criteria as it currently stands. Therefore, we invite you to submit a revised version of the manuscript that addresses the points raised during the review process.

We look forward to receiving your revised manuscript.

Kind regards,

Salvatore Lorenzo

Academic Editor

PLOS ONE

Journal Requirements:

   "No"

   "No" 

4. We note you have included a table to which you do not refer in the text of your manuscript. Please ensure that you refer to Table 1 in your text; if accepted, production will need this reference to link the reader to the Table.

Reviewers' comments:

Reviewer's Responses to Questions

**Comments to the Author**

1. Is the manuscript technically sound, and do the data support the conclusions?

Reviewer #1: Yes

2. Has the statistical analysis been performed appropriately and rigorously? 

Reviewer #1: N/A

3. Have the authors made all data underlying the findings in their manuscript fully available?

Reviewer #1: Yes

4. Is the manuscript presented in an intelligible fashion and written in standard English?

Reviewer #1: Yes

5. Review Comments to the Author

Reviewer #1: In this work, the authors study quantum walks and present a qualitative comparison between their behavior and some (known) results from decision theory. They analyze how different parameters associated with the initial state and the coin-flip operator affect the final distribution of the walker.

The main results presented in the manuscript seem to contribute to the literature and be technically correct. However, I do have some comments and clarifications that I would like the authors to address. Most of them are related to mathematical and physical concepts since they are my area of expertise.

1. To start, I would like to mention a few typos I identified across the manuscript.

- In line 78, “This serve” should read as “This serves.”

- In lines 148 and 149, it should be observed that the word scissors does not have a singular form. Then, “scissor” should be replaced by “scissors” or “a pair of scissors.”

- There is a missing comma at the end of Eq. (25).

- In line 501, “taking measurement from Eq (31) on the position bases” should read as “taking the measurement from Eq (31) at the locations.”

- In line 519, “taking measurement Eq (31)” should read as “taking the measurement from Eq (31).”

- In lines 654 and 656, “decision maker” should read as “decision-maker.”

- In line 656, “makes” should be replaced with “make.”

- In line 681, “results” should read as “result.”

3. At least the first time the term Hilbert space is mentioned (lines 160 and 161), the fact that the space is a complex Hilbert space should be emphasized. It does not help that the example mentioned in line 165 is the Euclidian (real) space.

4. In lines 164 and 165, it is said that <s_i| complex="" conjugate="" is="" of="" the="">. In reality, the former is the adjoint of the latter. In the case of finite vector spaces, the adjoint is the conjugate transpose.

5. The terms “basis” and “bases” are not properly used in the manuscript. Every time the authors refer to the element of a basis (for instance, |x>), they call it a basis. Meanwhile, the collection of all elements of a basis form a basis (and not a bases). In a vector space of dimension d over the real or complex numbers, an infinite bases (with each basis containing d elements) can be constructed. This problem appears in multiple places. A (possibly non-exhaustive) list of them is lines 162, 166, 167, 169, 170, 192, 351, 357, 362, and 413.

6. In lines 166 and 167, the sentence “Without loss of generality, we will use the term vector for the bases and states in the Hilbert space” is unnecessary. It is also slightly problematic in the sense that the correct statement would be: “Pure quantum states are represented by vectors in the Hilbert space.”

7. When talking about density matrices in lines 174 and 175, the authors state that it “is an ensemble of pure states.” This is not always the case because density matrices can also be associated with quantum systems entangled with other quantum systems via a partial trace. Because of it, even the term “statistical uncertainty” in line 179 may generate disagreement among physicists.

8. In line 180, when mentioning that the evolution of a system is unitary, it should be mentioned that the evolution of a closed system is unitary. In general, this is not the case for the dynamics of open quantum systems.

9. The form for the unitary U in line 182 only holds if H is time-independent. This should be explicitly mentioned.

10. In the subsection “Measurement and observable,” it should be stated that only projective measurements are being described. The most general measurement, not discussed there, is represented by a positive operator-valued measure (POVM).

11. In line 194, I believe the authors meant to write “pure state” instead of “quantum state.”

12. In line 196, when stating that a quantum state collapses, it is prudent to add between parenthesis “(or is updated)” since the word “collapse” might suggest specific interpretations about the measurement process and the ontology of quantum states.

13. In the subsection “The essential ingredient: quantum interference,” I suggest that the authors explicitly add the expression that defines q after Eq. (18). Moreover, I have an additional comment that does not need to be taken into consideration for the current manuscript but may help the authors in future works. More precisely, I want to briefly mention why I have issues with the approach they used to discuss interference in this subsection. First, to me, Eq. (18) can be read as a proof that quantum systems do not admit joint probability (even though I understood the specific way “joint probability” was defined in this work). Second, the current presentation might give the impression that interference is associated with measurements alone when, in fact, it is a characteristic of the dynamics of quantum systems. Observe that one of the main distinctions between classical statistical computation and quantum computation is the fact that, in the former, probabilities are directly manipulated. At the same time, in the latter, complex amplitudes are the quantities that can be manipulated.

14. In line 408, it should be clarified that although the “position space” under consideration is associated with a single spatial dimension, the dimension of the Hilbert space itself is infinite (since x runs over the integers).

15. In line 427, the phrase “incorporates both quantum and Kolmogorovian probabilities” seems to make more sense to me when referring to density matrices resulting from the statistical mixture of pure states (in fact, the rest of the paragraph suggests that this is what the authors had in mind). However, as already mentioned in this report, density matrices also arise from partial traves of composed entangled systems. In these cases, it is possible to argue that density matrices originating from partial traces do not necessarily have Kolmogorovian probabilities associated with them.

16. In line 453, the authors write that “the state propagates without deformation and independently along the left and right directions.” However, in the case under consideration, the equation for psi_R reads as psi_R(x,t) = - psi_R(x-1, t-1). Does the negative sign not have any effect on the dynamics? Could it be that the probability distribution of the system is what is not affected during the dynamics?

17. Maybe the authors could mention that the operator in Eq. (35) is generally a POVM. In the special case of projective measurements, {M_i} are orthogonal projectors, and the POVM reduces to a projection-valued measure (PVM).

18. In line 614, the sentence “This corresponds to a classical random walk evolving in parallel to a quantum walk” does not make much sense to me. I believe the authors mean that the resulting distribution corresponds, in a sense, to an interpolation between a quantum walk and a classical random walk. While the last sentence is correct, I would not say these two walks are “running in parallel.”

19. In line 617, the sentence “recognizing that the spin state at each x is equivalent to a qubit” can be removed because there is not much content to it. I would not say that the coin system is a “spin at each x.” Instead, it can be thought of as the spin of the system. Moreover, a qubit, by definition, is any two-dimensional quantum system. Then, a spin is a qubit.

20. The sentence “Fig 5 shows that a small value of p_m (0.05, 0.15) already gives a characteristic shape of a quantum random walk, which is very different from the classical random walk recovered for p_m = 1” in lines 638 to 640 does not seem to be correct. For values of p_m between 0.05 and 0.15, the distribution starts having its maximum value at zero. This shows that a bit of decoherence is enough for the quantum walk to approach a classical random walk, in accordance with the analysis of the variance in Fig. 6. Moreover, the authors used the term “quantum random walk.” For consistency across the text, I suggest the replacement of this phrase with “quantum walk.”

21. In lines 654 to 656, the conclusion that “if a decision maker constantly “checks” her/his state of mind, the behavior will shift towards a non-interference mode. This resonates with the evidence that continuous probing and self-assessment makes the decision maker more rational” seems quite surprising. In quantum computing, for instance, decoherence must be avoided at all costs to ensure quantum advantage can be achieved. Here, however, “decoherence” leads to advantages. This is one of the results that makes me question if quantum models are really necessary for decision theory. Can the authors comment on this?

22. In lines 680 to 682, the sentence “If the distance from the starting point and one end (say, left end) is a, and the distance from the other end is b, given infinite time, it is a standard results in classical diffusion theory that the probability that a classical particle hits the left end first is b/(a + b)” seems to be valid only if the coin used for the classical walk is fair. Is this the case?

23. In lines 742 to 745, the authors stated that “Behaviorally, rho can be interpreted as regulating an “internal time constraint”, for example to save computational resources, even when there is no explicit time pressure. The parameter t may be thought of as exogenous (e.g. a time limit imposed by the experimenter or the environment).” Because of the trade-off between rho and the total time of the quantum walk, would it be possible to, alternatively, fix rho and define an effective time for the dynamics? The effective time would be a function of the internal and external time constraints.

24. In Supplemental Information 1, the authors write that “There is one main difference between the standard theory of quantum measurements and QDT: the former considers measurements over passive systems, so there is no preferred quantum state, and any basis can be employed. In QDT, the decision maker is an active entity, represented by her own state of mind, that encodes her personal attributes/preferences.” I would like to comment that, even in quantum mechanics, there is a preferred basis in various scenarios. An example is the so-called computation basis used in quantum circuits and, more generally, in systems used for quantum control, like the coin system in a quantum walk.

25. Is the gain of replacing classical models for quantum ones only conceptual? If it is, one should evaluate if this shift is really worth it because quantum mechanics itself comes with many interpretational issues. What is the position of the authors on this aspect?

I hope the comments/questions presented here help improve the work.</s_i|>

6. PLOS authors have the option to publish the peer review history of their article (what does this mean?). If published, this will include your full peer review and any attached files.

Reviewer #1: No

---

## [Author Response · Author response to Decision Letter 0]

8 Aug 2022

We are grateful and thank the academic editor Salvatore Lorenzo and the reviewer for their thoughtful comments and suggestions that helped indeed improve the manuscript. We have edited the manuscript to address their concerns. Below is our response to each point raised by reviewer #1. We locate the relevant changes in the manuscript with line numbers appearing in the “Revised Manuscript with Track Changes.pdf”. We also mention the other changes we have made in the section after responses to reviewer #1.

We believe that the manuscript is now suitable for publication in PLOS ONE.

Ming Chen (PhD candidate at ETH Zurich)

On behalf of all authors.

Response to Reviewer #1:

In this work, the authors study quantum walks and present a qualitative comparison between their behavior and some (known) results from decision theory. They analyze how different parameters associated with the initial state and the coin-flip operator affect the final distribution of the walker.

The main results presented in the manuscript seem to contribute to the literature and be technically correct. However, I do have some comments and clarifications that I would like the authors to address. Most of them are related to mathematical and physical concepts since they are my area of expertise.

1. To start, I would like to mention a few typos I identified across the manuscript.

- In line 78, “This serve” should read as “This serves.”

- In lines 148 and 149, it should be observed that the word scissors does not have a singular form. Then, “scissor” should be replaced by “scissors” or “a pair of scissors.”

- There is a missing comma at the end of Eq. (25).

- In line 501, “taking measurement from Eq (31) on the position bases” should read as “taking the measurement from Eq (31) at the locations.”

- In line 519, “taking measurement Eq (31)” should read as “taking the measurement from Eq (31).”

- In lines 654 and 656, “decision maker” should read as “decision-maker.”

- In line 656, “makes” should be replaced with “make.”

- In line 681, “results” should read as “result.”

Response: All the above mentioned typos have been corrected in the manuscript.

2. At least the first time the term Hilbert space is mentioned (lines 160 and 161), the fact that the space is a complex Hilbert space should be emphasized. It does not help that the example mentioned in line 165 is the Euclidian (real) space.

Response: We have deleted the sentence “a Hilbert space is the Euclidean vector space” (line 167). “Complex” is mentioned in line 174.

3. In lines 164 and 165, it is said that . In reality, the former is the adjoint of the latter. In the case of finite vector spaces, the adjoint is the conjugate transpose.

Response: We have corrected this point in the manuscript (line 166).

4. The terms “basis” and “bases” are not properly used in the manuscript. Every time the authors refer to the element of a basis (for instance, |x>), they call it a basis. Meanwhile, the collection of all elements of a basis form a basis (and not a bases). In a vector space of dimension d over the real or complex numbers, an infinite bases (with each basis containing d elements) can be constructed. This problem appears in multiple places. A (possibly non-exhaustive) list of them is lines 162, 166, 167, 169, 170, 192, 351, 357, 362, and 413.

Response: This has been corrected in the manuscript (lines 163, 164, 165, 172, 173, 174, 202, 364, 370, 375, 422, 427, 430, 521, 646, 647, 777, 898, 899).

5. In lines 166 and 167, the sentence “Without loss of generality, we will use the term vector for the bases and states in the Hilbert space” is unnecessary. It is also slightly problematic in the sense that the correct statement would be: “Pure quantum states are represented by vectors in the Hilbert space.”

Response: This has been corrected in the manuscript (lines 168, 169, 170).

6. When talking about density matrices in lines 174 and 175, the authors state that it “is an ensemble of pure states.” This is not always the case because density matrices can also be associated with quantum systems entangled with other quantum systems via a partial trace. Because of it, even the term “statistical uncertainty” in line 179 may generate disagreement among physicists.

Response: We have added the expressions “quantum systems entangled with other quantum systems via a partial trace” (lines 179, 180) and “the minimum knowledge about the quantum system” (lines 185, 186).

7. In line 180, when mentioning that the evolution of a system is unitary, it should be mentioned that the evolution of a closed system is unitary. In general, this is not the case for the dynamics of open quantum systems.

Response: This has been corrected in the manuscript (lines 187, 192, 193, 194, 195).

8. The form for the unitary U in line 182 only holds if H is time-independent. This should be explicitly mentioned.

Response: This has been corrected in the manuscript (lines 190, 191).

9. In the subsection “Measurement and observable,” it should be stated that only projective measurements are being described. The most general measurement, not discussed there, is represented by a positive operator-valued measure (POVM).

Response: This has been corrected in the manuscript and added a reference (lines 199, 200)

10. In line 194, I believe the authors meant to write “pure state” instead of “quantum state.”

Response: This has been corrected in the manuscript (lines 204, 206).

11. In line 196, when stating that a quantum state collapses, it is prudent to add between parenthesis “(or is updated)” since the word “collapse” might suggest specific interpretations about the measurement process and the ontology of quantum states.

Response: This has been added in the manuscript (line 207).

12. In the subsection “The essential ingredient: quantum interference,” I suggest that the authors explicitly add the expression that defines q after Eq. (18). Moreover, I have an additional comment that does not need to be taken into consideration for the current manuscript but may help the authors in future works. More precisely, I want to briefly mention why I have issues with the approach they used to discuss interference in this subsection. First, to me, Eq. (18) can be read as a proof that quantum systems do not admit joint probability (even though I understood the specific way “joint probability” was defined in this work). Second, the current presentation might give the impression that interference is associated with measurements alone when, in fact, it is a characteristic of the dynamics of quantum systems. Observe that one of the main distinctions between classical statistical computation and quantum computation is the fact that, in the former, probabilities are directly manipulated. At the same time, in the latter, complex amplitudes are the quantities that can be manipulated.

Response: The expression defining q has been added (line 246). We also thank the referee for his/her thoughtful suggestions. Indeed, how to define the concept of a joint probability is delicate and a matter of continuing controversies. Let us note for the interest of this referee that Yukalov and one of the authors have suggested a definition of quantum joint probabilities by introducing composite events in multichannel measurements (see [1]).

13. In line 408, it should be clarified that although the “position space” under consideration is associated with a single spatial dimension, the dimension of the Hilbert space itself is infinite (since x runs over the integers).

Response: This has been corrected in the manuscript (lines 421, 422).

14. In line 427, the phrase “incorporates both quantum and Kolmogorovian probabilities” seems to make more sense to me when referring to density matrices resulting from the statistical mixture of pure states (in fact, the rest of the paragraph suggests that this is what the authors had in mind). However, as already mentioned in this report, density matrices also arise from partial traces of composed entangled systems. In these cases, it is possible to argue that density matrices originating from partial traces do not necessarily have Kolmogorovian probabilities associated with them.

Response: We have reformulated it in lines 441, 442, 443.

15. In line 453, the authors write that “the state propagates without deformation and independently along the left and right directions.” However, in the case under consideration, the equation for psi_R reads as psi_R(x,t) = - psi_R(x-1, t-1). Does the negative sign not have any effect on the dynamics? Could it be that the probability distribution of the system is what is not affected during the dynamics?

Response: This is correct, it is the probability distribution that is unaffected, not the state. We have corrected this point (lines 468, 469).

16. Maybe the authors could mention that the operator in Eq. (35) is generally a POVM. In the special case of projective measurements, {M_i} are orthogonal projectors, and the POVM reduces to a projection-valued measure (PVM).

Response: We have added this point in the manuscript and added a reference (lines 500, 501, 502, 503).

17. In line 614, the sentence “This corresponds to a classical random walk evolving in parallel to a quantum walk” does not make much sense to me. I believe the authors mean that the resulting distribution corresponds, in a sense, to an interpolation between a quantum walk and a classical random walk. While the last sentence is correct, I would not say these two walks are “running in parallel.”

Response: We have removed the sentence (line 635).

18. In line 617, the sentence “recognizing that the spin state at each x is equivalent to a qubit” can be removed because there is not much content to it. I would not say that the coin system is a “spin at each x.” Instead, it can be thought of as the spin of the system. Moreover, a qubit, by definition, is any two-dimensional quantum system. Then, a spin is a qubit.

Response: We have removed this sentence (lines 637, 638).

19. The sentence “Fig 5 shows that a small value of p_m (0.05, 0.15) already gives a characteristic shape of a quantum random walk, which is very different from the classical random walk recovered for p_m = 1” in lines 638 to 640 does not seem to be correct. For values of p_m between 0.05 and 0.15, the distribution starts having its maximum value at zero. This shows that a bit of decoherence is enough for the quantum walk to approach a classical random walk, in accordance with the analysis of the variance in Fig. 6. Moreover, the authors used the term “quantum random walk.” For consistency across the text, I suggest the replacement of this phrase with “quantum walk.”

Response: We have reformulated this sentence, following the suggestions of the referee (lines 659, 660, 661),

20. In lines 654 to 656, the conclusion that “if a decision maker constantly “checks” her/his state of mind, the behavior will shift towards a non-interference mode. This resonates with the evidence that continuous probing and self-assessment makes the decision maker more rational” seems quite surprising. In quantum computing, for instance, decoherence must be avoided at all costs to ensure quantum advantage can be achieved. Here, however, “decoherence” leads to advantages. This is one of the results that makes me question if quantum models are really necessary for decision theory. Can the authors comment on this?

Response: This is an interesting comment by the referee. This illustrates that the goals and metric of success in quantum computing and in quantum decision theory are different. Quantum computing aims at achieving quantum supremacy, i.e., developing calculations that are much faster than classical computations. For this, it relies fundamentally on quantum coherence and interferences. In contrast, quantum decision theory aims at describing and explaining real imperfect human decision making with as few assumptions and adjustable parameters as possible. In this role, quantum decision theory succeeds beautifully as reviewed in the manuscript. And with respect to the fact that multiple self-assessment and measurements make the decision more rational, recovering the objective probability, turns out to be a great prediction of the quantum walk approach, which is born out by experiments on real humans. In this sense, one could state that real humans have too much coherence and entanglement in their state of mind and this creates biased subjective assessment of probabilities and a host of puzzles and fallacies.

21. In lines 680 to 682, the sentence “If the distance from the starting point and one end (say, left end) is a, and the distance from the other end is b, given infinite time, it is a standard results in classical diffusion theory that the probability that a classical particle hits the left end first is b/(a + b)” seems to be valid only if the coin used for the classical walk is fair. Is this the case?

Response: Yes, you are right and we write it explicitly in line 702. 

22. In lines 742 to 745, the authors stated that “Behaviorally, rho can be interpreted as regulating an “internal time constraint”, for example to save computational resources, even when there is no explicit time pressure. The parameter t may be thought of as exogenous (e.g. a time limit imposed by the experimenter or the environment).” Because of the trade-off between rho and the total time of the quantum walk, would it be possible to, alternatively, fix rho and define an effective time for the dynamics? The effective time would be a function of the internal and external time constraints.

Response: It is definitely possible to do so, however, we thought it would be more intuitive for the decision-making community to see an internal parameter attached to the decision-maker, rather than an effective time for each decision-maker.

23. In Supplemental Information 1, the authors write that “There is one main difference between the standard theory of quantum measurements and QDT: the former considers measurements over passive systems, so there is no preferred quantum state, and any basis can be employed. In QDT, the decision maker is an active entity, represented by her own state of mind, that encodes her personal attributes/preferences.” I would like to comment that, even in quantum mechanics, there is a preferred basis in various scenarios. An example is the so-called computation basis used in quantum circuits and, more generally, in systems used for quantum control, like the coin system in a quantum walk.

Response: We thank the referee for this precision and we have deleted footnote 1 in “S1_Appendix.pdf”, accordingly.

24. Is the gain of replacing classical models for quantum ones only conceptual? If it is, one should evaluate if this shift is really worth it because quantum mechanics itself comes with many interpretational issues. What is the position of the authors on this aspect?

Response: This is a fair question and we suggest the following considerations to the referee. First, it is true that finding a classical-quantum boundary in decision-making is very hard and its use is difficult to assess at the present stage of knowledge. For example, given the stochastic outcomes of human choices, one may explain it with random preference approaches [2] or quantum approaches (quantum measurements result in inherent stochasticity). The axioms of both approaches are not comparable and it is very hard to draw a boundary between them (section 3.2.4 of the paper [3]).

Having said that, we still think quantum decision models are necessary and useful as a foundation of any other decision theories. To achieve the same explanatory power, a quantum model is simpler than a classical model. Many theories solve one or several specific problems by proposing new assumptions or amending some of von Neumann and Morgenstern’s axioms and Savage’s Axioms. A quantum model provides a more general framework with Hilbert space and quantum probability, where no “anomalies” and “paradoxes” arise. A few examples in our paper demonstrate its simplicity and explanatory power: it captures normative or rational decision behaviors in the special case of full decoherence without the need of “fast” and “slow” systems; it naturally accounts for conjunction and disjunction fallacies as the result of the nature of quantum probability. 

Furthermore, theories based on a quantum framework are proved to outperform the classical ones in data calibration [3].

Finally, let us mention that the third author and Yukalov have investigated this question in great length, see the series of papers at https://er.ethz.ch/media/publications/social-systems-finance/decision_theory.html that should be read from bottom to top.

Other changes:

1. We have removed the part of sentence “d is normally finite in implementations of decision theories” as, in quantum walks, the position space has infinite dimensions. (line 162)

2. We have merged the paragraphs “Mixed state” into “Quantum state” as mix states are also quantum states. (line 178)

3. We have changed the expression in line 483 “time-depend states” to “time-dependent solutions” 

Reference:

[1] V. l. Yukalov and D. Sornette, "Quantum probabilities of composite events in quantum measurement with multimode states," Laser Physics, vol. 23, no. 10, 2013. 

[2] C. Manski and D. McFadden, Eds., Structural Analysis of Discrete Data with Econometric Applications, Cambridge: MIT Press, 1981, pp. 198-272.

[3] G. Ferro, T. Kovalenko and D. Sornette, "Quantum decision theory augments rank-dependent expected utility and Cumulative Prospect Theory," Journal of Economic Psychology, vol. 86, no. 102417, 2021.

---

## [Editor Report · Decision Letter 1]

11 Aug 2022

On the use of discrete-time quantum walks in decision theory

PONE-D-22-13566R1

Dear Dr. Chen,

We’re pleased to inform you that your manuscript has been judged scientifically suitable for publication and will be formally accepted for publication once it meets all outstanding technical requirements.

Kind regards,

Salvatore Lorenzo

Academic Editor

PLOS ONE
---

## [Editor Report · Acceptance letter]

18 Aug 2022

PONE-D-22-13566R1 

On the use of discrete-time quantum walks in decision theory 

Dear Dr. Chen:

I'm pleased to inform you that your manuscript has been deemed suitable for publication in PLOS ONE. Congratulations! Your manuscript is now with our production department. 

Kind regards, 

on behalf of

Dr. Salvatore Lorenzo 

Academic Editor

PLOS ONE